# Dissection of intercellular communication using the transcriptome-based framework ICELLNET

Floriane Noël [1,2,3,7], Lucile Massenet-Regad [1,4,7], Irit Carmi-Levy[2,3], Antonio Cappuccio[2,3], Maximilien Grandclaudon[2,3], Coline Trichot[1,2,3], Yann Kieffer [2,5], Fatima Mechta-Grigoriou [2,5] & Vassili Soumelis [1,2,3,6 ✉]

Cell-to-cell communication can be inferred from ligand–receptor expression in cell transcriptomic datasets. However, important challenges remain: global integration of cell-to-cell communication; biological interpretation; and application to individual cell population transcriptomic profiles. We develop ICELLNET, a transcriptomic-based framework integrating: 1) an original expert-curated database of ligand–receptor interactions accounting for multiple subunits expression; 2) quantification of communication scores; 3) the possibility to connect a cell population of interest with 31 reference human cell types; and 4) three visualization modes to facilitate biological interpretation. We apply ICELLNET to three datasets generated through RNA-seq, single-cell RNA-seq, and microarray. ICELLNET reveals autocrine IL-10 control of human dendritic cell communication with up to 12 cell types. Four of them (T cells, keratinocytes, neutrophils, pDC) are further tested and experimentally validated. In summary, ICELLNET is a global, versatile, biologically validated, and easy-to-use framework to dissect cell communication from individual or multiple cell-based transcriptomic profiles.

[1] Université de Paris, INSERM U976, Equipe labellisée par la Ligue Nationale contre le Cancer, F-75006 Paris, France. [2] Institut Curie, 26 rue d'Ulm, Paris, France. [3] INSERM U932, Immunity and Cancer, PSL Research University, Paris, France. [4] Université Paris-Saclay, Saint Aubin 91190, France. [5] INSERM U830, Stress and Cancer Laboratory, Equipe labellisée par la Ligue Nationale contre le Cancer, PSL Research University, Paris, France. [6] AP-HP, Hôpital Saint-Louis, Département d'Immunologie-Histocompatibilité, Paris 75010, France. [7] These authors contributed equally: Floriane Noël, Lucile Massenet-Regad. ✉email: vassili.soumelis@aphp.fr

Cell-to-cell communication is at the basis of the higher order organization observed in tissues, organs, and organisms, at steady-state and in response to stress. It involves a messenger or sender cell, which transmits information signals to a receiving or target cell. Information is generally coded in the form of a chemical molecule that is sensed by the target cell through a cognate receptor. Multiple cells or cell types communicating with each other form cell communication networks.

In mammalian organisms, endocrine communication involves cells that may be at very distant anatomical sites. However, cell communication also takes place locally through cell-to-cell contacts, or through inflammatory molecules. Cytokines and other mediators can be involved in distant as well as local communication[1–3]. Hence, when deciphering cell-to-cell communication, one should account for potential signals coming both from spatially proximal and distal cells.

Most studies in the past decades have focused on a limited number of communication molecules in a given anatomical site or physiological process. The availability of large-scale transcriptomic datasets from several cell types, tissue locations, and cell activation states, opened the possibility of reconstructing cell-to-cell interactions based on the expression of specific ligand–receptor pairs on sender and target cells, respectively. Many of them exploit single-cell RNA-seq datasets to infer communication between groups of cells within the same dataset[4–7]. Despite leading to interesting and often innovative hypotheses[4,6,8], these methods do not integrate putative signals that may come from more distant cells. Also, they cannot be applied to bulk transcriptomic data derived from a given cell population. Such datasets are numerous in public databases, and can be a source of novel insights into how each cell type may send or receive communication signals.

Another important aspect when inferring cell-to-cell communication is the use of databases of ligand–receptor interactions. Some are very broad with over 2000 ligand–receptor pairs[9], but lack systematic manual or expert curation, which may impact the quality and biological relevance of the annotation. Others include lower numbers of ligand–receptor pairs and provide manually curated information from the literature[4,10], without necessarily providing systematic combinatorial rules for the association of protein subunits into multimeric ligands or receptors.

The last point relates to the granularity that is structuring the biological information into families and subfamilies of functionally and structurally related molecules. We only found one tool that provides a classification into four families of communication molecules[5], while suffering from other limitations in particular the lack of manual curation.

In this study we develop ICELLNET, a versatile computational framework to infer cell-to-cell communication from a wide range of bulk and single-cell transcriptomic datasets. Each family of communication molecules is expert curated and organized into biologically relevant sub-families. ICELLNET offers an array of visualization tools in order to facilitate biological interpretation and discoveries. We provide applications to public datasets generated using different technologies and our own original transcriptomic datasets, in non-immune (tumor fibroblasts) and immune cell types. Experimental validation of ICELLNET-derived predictions demonstrated IL-10 control of human dendritic cell communication.

## Results

### Expert-curated database of ligand–receptor interactions.
In order to globally reconstruct cell communication networks, we curated a comprehensive database of ligand–receptor interactions from the literature[3,11,12] and public databases[10,13]. An expert manual curation was performed based on a rigorous literature screening of original articles, applying the following criteria: 1) robustness of the findings, 2) consistency with international classifications and nomenclature, 3) experimental validation of the functionality of the ligand–receptor interaction (see Methods). We also used consensus reviews from leaders in the field, in particular for cytokines[3,14–16]. This helped solving controversies on how to classify some molecules, which have features of different molecular families. We did not include putative interactions based on protein–protein interaction predictions, as it is done in some other databases[9]. This led to the integration of 380 ligand–receptor interactions into the ICELLNET database (Supplementary Data 1). Whenever relevant, we took into account the multiple subunits of the ligands and the receptors (Fig. 1a). Interactions were classified into 6 major families of communication molecules, with a strong emphasis on inflammatory and immune processes: Growth factors, Cytokines, Chemokines, Immune Checkpoints, Notch signaling, and Antigen binding (Fig. 1b and Supplementary Data 1). Other families such as hormones or adhesion molecules were more scarcely represented. In order to simplify the subsequent graphical visualization, these were grouped as other in our current classification (Fig. 1b and Supplementary Data 1).

Cytokine–receptor pairs were mapped in an exhaustive manner, by exploiting a series of reference articles and consensus classifications. They represent 50% of the total interactions included in the database (194 interactions), and were further classified into 7 sub-families according to structural protein motifs: type 1 cytokines, type 2 cytokines, IL-1 family, IL-17 family, TNF family, TGF-ß family, and RTK cytokines[3,14–16] (Fig. 1c).

This database is integrating information on both multiple subunits of ligands and receptors, and a classification into molecular families/subfamilies.

### Development of a computational pipeline to dissect intercellular communication.
In the ICELLNET framework, we developed an automatized tool in R script to infer communication between multiple cell types by integrating: 1) prior knowledge on ligand–receptor interactions (Fig. 1); 2) computation of a communication score between pairs of cells based on their transcriptomic profiles, and; 3) several visualization modes to guide results interpretation. Quantification of intercellular communication was achieved by scoring the intensity of each ligand–receptor interaction between two cell types from their expression profiles (Fig. 2).

From each transcriptomic profile, all genes or only differentially expressed genes could be used, and no filtering threshold was applied to gene expression. Taking advantage of the ICELLNET database, the genes coding for ligands/receptors were selected from all 380 interactions to compute the score, but it is also possible to restrict the database to specific families of molecules, depending on the biological question.

A unique feature and strength of ICELLNET is its ability to infer cell-to-cell communication even from an individual cell population-based transcriptome of interest (the "central cell"). The user should define this cell type of interest, and provide its transcriptional profile, which for example may be selected from different primary cell subsets, or from the same cell type cultured in different biological conditions (Fig. 2 top-left). ICELLNET separately considers other cell types with known transcriptomic profiles (hereafter called « partner cells ») that can connect to the central cell. These can be cell types coming from the same dataset as the central cell, or from any other transcriptomic dataset. The ICELLNET pipeline has integrated reference transcriptomic

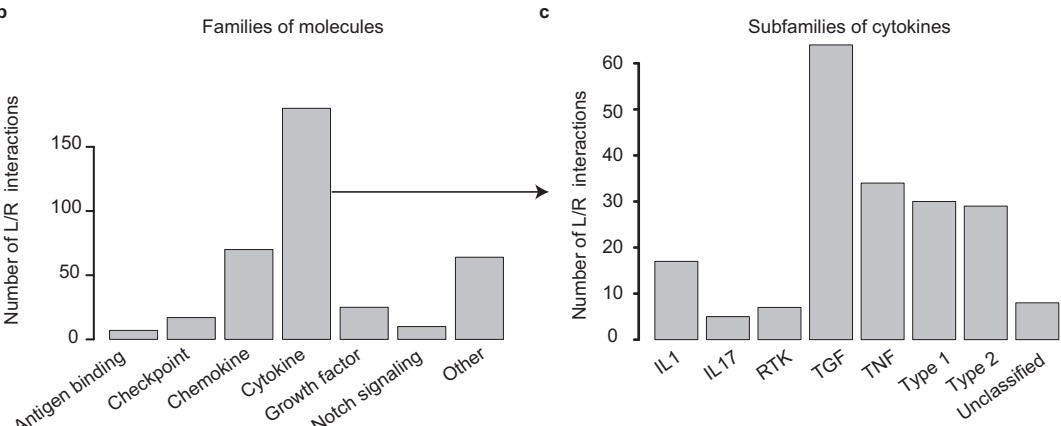

**a**

| Ligand 1 | Ligand 2 | Receptor 1 | Receptor 2 | Family | Subfamily | PubMed ID |
|---|---|---|---|---|---|---|
| CCL11 | | CXCR3 | | Chemokine | | 12884299 |
| ICOSLG | | ICOS | | Checkpoint | | 14599850 ; 23954143 |
| TSLP | | IL7R | CRLF2 | Cytokine | Type 1 | 11418668 |
| IFNL1 | | IL10RB | IFNLR1 | Cytokine | Type 2 | 12469119 |
| INHBA | INHBB | ACVR1B | ACVR2A | Cytokine | TGF | 8622651; 8721982 |

**Fig. 1 Structure of the ligand–receptor database. a** Extract of the ligand–receptor database. The database is structured as the following scheme: the 4 first columns correspond to gene symbol of each subunit of interacting ligand and receptor, the next two columns state the classification into families and/or subfamilies of molecules, and the last column gives the source for manual curation (PubMed ID). **b** Histogram displaying the number of interactions classified in each family of communication molecules included in the database (antigen binding, checkpoint, chemokine, cytokine, growth factor, notch signaling). **c** Histogram displaying the number of interactions classified in the different defined subfamilies of cytokines: interleukin 1 family, interleukin 17 family, receptor tyrosine kinase (RTK) family, transforming growth factor beta (TGF) family, tumor necrosis factor family (TNF), type 1 cytokine family, and type 2 cytokine family, and unclassified cytokines.

profiles by using the Human Primary Cell Atlas[17,18]. This public dataset includes transcriptomic profiles of 31 human cell types including immune cells, stromal cells, neural cells, and tissue-specific cell types, all generated with the same Affymetrix technology (Fig. 2 top-right). Human Primary Cell Atlas has been downloaded and added to ICELLNET framework, in order to be used as reference transcriptomic profiles of partner cell types (Supplementary Data 2). The user always has the possibility of providing original transcriptional profiles for central and partner cells, or to use the reference transcriptomic profiles included in ICELLNET R package to represent putative « partner » cells.

**Establishment of a score to assess the communication between cells.** From the transcriptomic profiles, we selected the genes coding for the ligands and the receptors in our database. We designed the tool to enable the user to focus on the ligands and/or receptors that are differentially expressed between conditions of study, or to use the entire ligand–receptor database to compute the communication score.

Since cell-to-cell communication is directional, we considered ligand expression from the central cell, and receptor expression from the partner cells in order to assess outward communication. Conversely, we then selected receptor expression from the central cell, and ligand expression from partner cells in order to assess inward communication (Fig. 2 middle). For each gene, expression levels were scaled by maximum of gene expression in the dataset, in order to avoid a communication score predominantly driven by highly expressed genes. Indeed, bioactivity of communication molecules varies a lot. Some cytokines, such as IL-12 and IL-4, are very bioactive at low concentrations, and often expressed at very

low levels both in transcript and protein. Conversely, many chemokines are produced at much higher levels, without necessarily having a higher bioactivity. Not scaling the data before inferring a communication score would systematically favor a few highly expressed molecules, and would not allow detecting the contribution of important molecules expressed at much lower levels. In the ICELLNET framework, quantification of intercellular communication consists of scoring the intensity of each ligand–receptor interaction between two cell types with known expression profiles. Whenever relevant, we took into account multiple ligand subunits, or receptor chains, using logical rules to impose their co-expression in order to consider functionality. The score of an individual ligand–receptor interaction was computed as the product of their expression levels by the respective source (central) and target (partner) cell. Whenever a communication molecule (ligand or receptor or both) was not expressed by a cell, the score of this particular interaction was set to zero. Individual scores were then combined into a global metric assessing the overall exchange of information between the cell types of interest (Fig. 2 middle), defining a global communication score. ICELLNET provides a matrix summarizing all global communication scores as an output of these analytical steps.

**ICELLNET offers different graphical representations allowing multiple layers of interpretation.** ICELLNET generates a large quantity of data and scores, which are complex to interpret and analyze. In order to facilitate hypothesis generation, three graphical representations were implemented to help visualize and interpret the results (Fig. 2 bottom). The first representation allows the visualization of intercellular communication networks

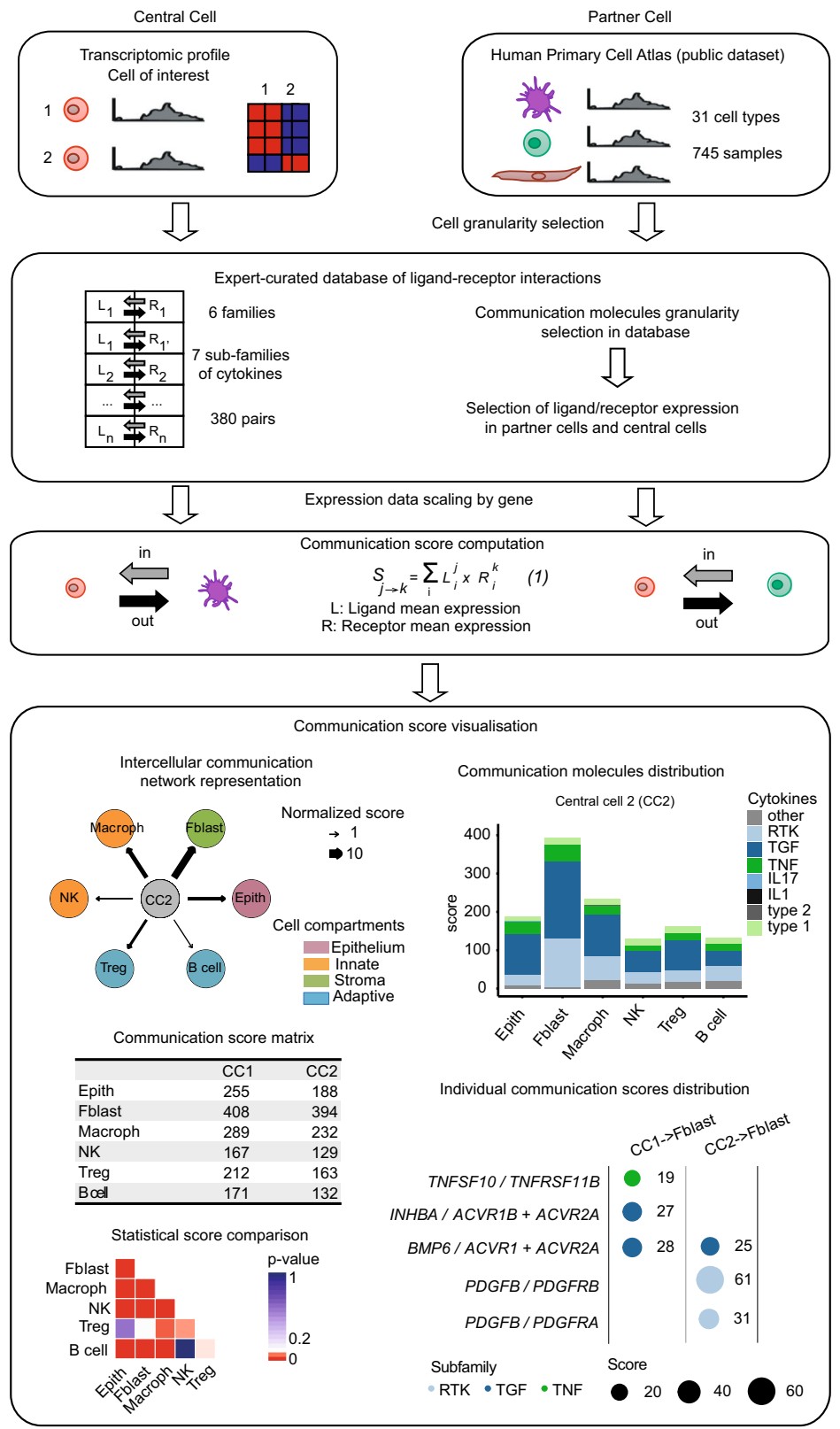

in directed connectivity maps. In these graphs, nodes represent cell types, the width of the edges connecting two cell types is proportional to their global communication score and the arrows indicate the direction of communication. The second visualization mode breaks down the global scores into the contribution of specific molecular families through a barplot representation. This allows the identification of patterns of co-expressed molecules

from the same family, potentially contributing to coordinated biological functions. We implemented statistical analyses of the scores (see Methods) to evaluate the robustness of the differences between scores. The resulting *p*-values can be visualized as an additional heatmap. The third representation displays the highest contributing ligand–receptor pairs to the communication score within a given channel in a balloon plot. This enables the

**Fig. 2 ICELLNET pipeline to study intercellular communication from cell transcriptional profiles.** (top) Selection of transcriptomic profile of cell population of interest (central cell) and of other cells used to infer communication with among a public dataset (partner cells) (Supplementary Data 2). (middle) Genes corresponding to ligands/receptors included in the database are selected, scaled by maximum of gene expression for each gene, and used to compute a communication score between two cell types. (bottom) Graphical layers used to dissect intercellular communication scores between the central cell (CC) and partner cells (Macroph: macrophage, Fblast: fibroblast, Epith: epithelial cell, B cell, Treg, NK: natural killer cell): global communication network intensity assessment, contribution of each family or subfamily of molecules to the communication scores, statistical difference assessment of the communication scores from the same central cell to the different partner cell types, and plots of specific interactions most contributing to communication scores. Some schematic art pieces were used and modified from Servier Medical Art, licensed under a Creative Common Attribution 3.0 Generic License. http://smart.servier.com/.

identification of specific interactions that may drive the global intercellular communication. Thus, the ICELLNET framework is a powerful tool to assess intercellular communication with different visualization modes that can be helpful to dissect underlying mechanisms and extend biological knowledge and understanding.

**Application of ICELLNET to study human breast cancer-associated fibroblasts.** Cancer-associated fibroblasts (CAFs) are stromal cells localized in the tumor microenvironment that are known to enhance tumor phenotypes, notably cancer cell proliferation, and inflammation. Recently, four subsets of CAFs have been identified and characterized in the context of previously untreated Luminal and Triple Negative Breast Cancer (TNBC)[19]. Notably, two subsets of CAFs named CAF-S1 and CAF-S4 specifically accumulated in TNBC microenvironment and CAF-S1 was associated with an immunosuppressive microenvironment. This study raised important questions about the regulatory mechanisms involved, in particular the role of cell-to-cell communication. Using the available transcriptional profiles of CAF-S1 and CAF-S4 in TNBC (Fig. 3a), we applied the ICELLNET pipeline to reconstruct the intercellular communication network with 14 other cell types potentially localized in the tumor microenvironment (TME) (Fig. 3b and Supplementary Data 3). The partner cells were selected from Human Primary Cell Atlas and included innate immune cells (monocytes, macrophages, pDC, DC1, DC2, NK cells, neutrophils), adaptive immune cells (CD4$^+$ T cells, CD8$^+$ T cells, Tregs, B cells), epithelial and stomal cells (fibroblasts and endothelial cells). In order to assess the global intercellular communication, we first used the network graphical visualization. This strongly suggested that CAF-S1 has a greater communication potential than CAF-S4 (Fig. 3b) to interact with other TME components. The rescaled communication scores were higher for CAF-S1 as compared to CAF-S4, and the differences were statistically significant for epithelial cells (score CAF S1 > Epith = 6, score CAF-S4 > Epith = 3, $p$ value < 0.1), endothelial cells (score CAF-S1 > Endoth = 7, score CAF-S4 > Endoth = 4, $p$ value < 0.1), plasmacytoid dendritic cells (score CAF-S1 > pDC = 6, score CAF-S4 > pDC = 4, $p$ value < 0.1) and B cells (score CAF-S1 > B cells = 3, score CAF-S4 > B cells = 1, $p$ value < 0.1) (Fig. 3b, c and Supplementary Data 3).

**CAF-S4 uses specific communication channels to interact with the TME components.** We focused on the biological composition of the score, to identify families of molecules highly involved in CAFs communication with the selected cells. We selected 4 partner cell types: epithelial cells, fibroblasts, Tregs and B cells. Using the barplot representation, we looked for differences between CAF-S1 and CAF-S4 in terms of genes coding for families of communication molecules. We found that genes coding for communication molecules inducing Notch signaling were specifically expressed by CAF-S4 to communicate with other cells (Fig. 3c). Looking at individual communication interactions

between CAF subsets and Tregs demonstrated that gene coding for JAG1 protein was only expressed by CAF-S4 to interact with NOTCH receptors (NOTCH1 and NOTCH2 genes expressed), and thus potentially having a role in activating the Notch signaling pathway (Fig. 3d and Supplementary Data 3). For both CAF subsets, the barplot representation indicated that cytokines–receptors interactions were highly contributing to the global communication scores compared to other families of molecules (Fig. 3c). This observation led us to focus on cytokine-mediated communication using the ICELLNET pipeline (Fig. 3e). By considering only cytokine–receptor interactions, the CAFs appear to communicate more with other fibroblasts compared to other cell types with a significant $p$-value (Fig. 3e, Supplementary Fig. 1a). Also, this approach highlighted that RTK cytokines, and notably PDGFB coding for PDGF, were preferentially expressed by CAF-S4 compared to CAF-S1 (Fig. 3e, Supplementary Fig. 1b and Supplementary Data 3). We also applied ICELLNET pipeline to study inward communication between the partner cells and the CAF subsets, which revealed no difference between CAF-S1 and CAF-S4 in term of communication score intensities but also in terms of the families of molecules involved in communication (Supplementary Fig. 2). Thus, the ICELLNET framework allowed us to identify specific communication channels revealing potential interactions between CAF-S4 and TME components.

**Lupus nephritis cell–cell communication network inferred from single-cell RNA-seq datasets using ICELLNET.** Single-cell technologies are now largely employed in various biological fields to better characterize immune cell diversity and cell phenotypes. They also offer insightful datasets to reconstruct cell–cell interactions between different cell populations from the same sample or tissue. We applied ICELLNET to a published single-cell dataset of immune cells from lupus nephritis patients[20]. This dataset included several immune cell subpopulations of T and B lymphocytes, but also natural killer cells, macrophages, and dendritic cell populations[20]. We represented those cells into a Uniform Manifold Approximation and Projection (UMAP) (Fig. 4a). We decided to focus on the potential communication between the conventional dendritic cell (cDC) cluster (CM3) and two clusters of T cells, CT0a and CT3b, which respectively refer to effector memory CD4$^+$ T cells and TFH-like cells according to the original study[20] (Fig. 4b). Because of sparsity and drop-out that are inherent to single-cell data, we computed the average gene expression profile for each cluster. Communication scores were then computed with cluster's mean expression profiles as input. The communication score between CM3 cluster and CT3b was higher than the score from CM3 to CT0a cluster (score CM3 > CT3b = 1527, score CM3 > CT0a = 1123) (Fig. 4b and Supplementary Data 4). In particular, it showed higher communication potential for checkpoints, chemokine, and growth factors (Fig. 4b). From this, we highlighted specific interactions that most differed between the two communication scores, such as CD86/CD28 (92 vs 40 for CM3 > CT3b and CM3 > CT0a, respectively), CD86/CTLA4 (92 vs 14, respectively), PDCD1LG2/PDCD1 (72 vs

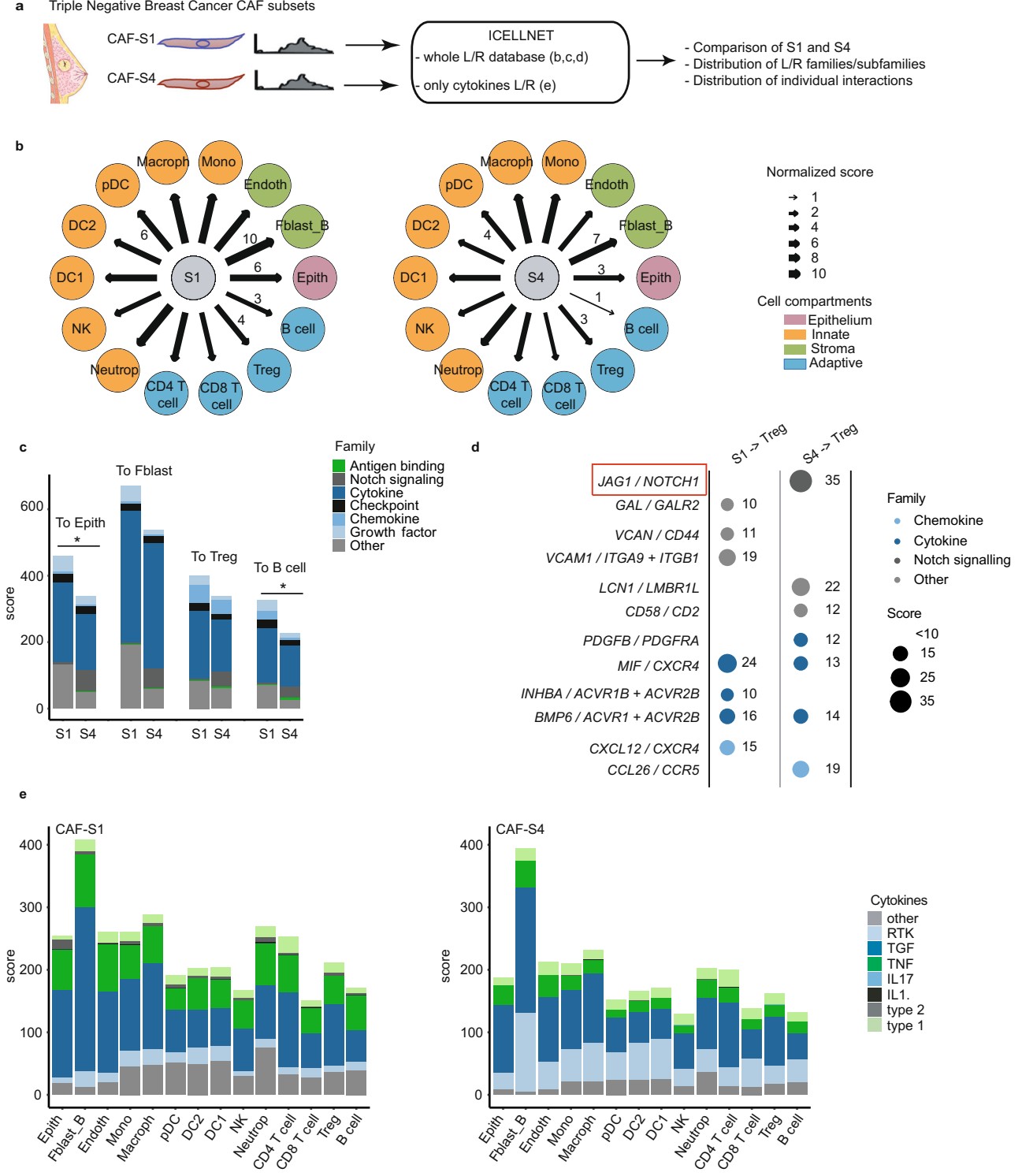

19), *CCL22/CCR4* (100 vs 39), or *VEGFA/FLT1* (21 vs 0) (Fig. 4c and Supplementary Data 4).

We then assessed the robustness of the communication scores by subsampling the clusters of interest (either CM3 cluster or CT0a and CT3b clusters). We randomly selected cells (number defined by the size of the clusters) from the clusters and computed again communication scores to assess variability (Fig. 4d). We observed that the variability of the communication score (measured using standard deviation) was anti-correlated with the percentage of cells used from the cluster, ranging from 37 and 59 (value of SD, for CT0a and CT3b communication, respectively) for 90% of the cluster cell to 146 and 215 (value of SD, for CT0a and CT3b communication, respectively) for 10% for CM3 subsampling (Fig. 4d and Supplementary Data 4). Similar results were observed for CT0a and CT3b cluster subsampling. This demonstrated the relevance of using the average of cluster gene expression to assess communication, rather than unique cells.

**Fig. 3 Dissection of intercellular communication between Triple-Negative breast cancer infiltrating CAF subsets and the tumor microenvironment.**
**a** Workflow of the analysis. **b** Connectivity maps describing outward communication from cancer associated fibroblasts CAF-S1 ($n = 6$ biologically independent samples) and CAF-S4 ($n = 3$ biologically independent samples) subsets to primary cells (Supplementary Data 2). The CAF subsets are considered as central cells and colored in gray. Primary cells are considered as partner cells (DC1 and DC2: conventional dendritic cell 1 and 2, pDC: plasmacytoid dendritic cell, Macroph: macrophage, Mono: monocyte, Endoth: endothelial cell, Fblast_B: breast fibroblast, Epith: epithelial cell, B cell, Treg, CD8 T cell, CD4 T cell, Neutrop: neutrophil, and NK: natural killer cell) and are colored depending on the cell compartment (green: stroma, orange: innate, blue: adaptive, pink: epithelium). The width of the edges corresponds to a global score combining the intensity of all the individual ligand/receptor interactions. A scale ranging from 1 to 10, corresponding to minimum and maximum communication scores, is shown in the legend. A selection of normalized scores is written directly on the network. **c** Barplot of communication score with contribution by families of communication molecules between CAF subsets ($n = 6$ biologically independent samples for CAF-S1, $n = 3$ biologically independent samples for CAF-S4) and a selection of partner cells. Significant differences are shown on the graph (two-sided wilcoxon test, and $p$ values are adjusted with Benjamin–Hochberg method, *$p$-value $\leq 0.1$). **d** Balloon plot of individual interaction scores between CAF subsets and Tregs. Only interactions with a score contribution above 10 to the score are displayed for clarity purpose. Two biologically interesting communication channels were highlighted by red boxes. **e** Barplot of communication score with contribution restricted to cytokines subfamilies between CAF subsets ($n = 6$ biologically independent samples for CAF-S1, $n = 3$ biologically independent samples for CAF-S4) and a selection of partner cells. Some schematic art pieces were used and modified from Servier Medical Art, licensed under a Creative Common Attribution 3.0 Generic License. http://smart.servier.com/ (**a**).

We also used the Human Primary Cell Atlas transcriptomic profiles of cDC to compute the communication scores between dendritic cells and the same T-cell clusters from the lupus nephritis dataset (CT0a and CT3b) (Supplementary Fig. 3a). Even if the cDC transcriptomic profile came from a different technology, it led to similar communication scores when comparing with the ones computed exclusively from the original single cell dataset (Fig. 4b, c, and Supplementary Fig. 3). The 122 ligand–receptor pairs contributing to the communication score in the single-cell dataset were also found using Human Primary Cell Atlas dataset (Supplementary Fig. 3c). The slight differences can be explained by the fact that the cells were not in the same biological state but also some genes were not captured with the Affymetrix technology. Nevertheless, it demonstrated that the Human Primary Cell Atlas can be very valuable to infer communication potential with a cell type that is not represented in a studied dataset.

**Application of ICELLNET to in vitro-generated immune cell states.** We then focused on a controlled in vitro system to assess whether ICELLNET would be able to predict cell–cell communication. We addressed the communication and underlying mechanisms in resting and activated DC. We generated an original microarray dataset, and applied ICELLNET to assess intercellular communication. LPS-activated human monocyte-derived DC produce two important autocrine cytokines, TNF and IL-10, which play a major role in regulating inflammation. We asked whether this could be mediated by modulating DC communication with partner cells. To this end, LPS-activated DC were cultured in the presence and absence of blocking antibodies (Abs) to the TNF and IL-10 receptors (aTNFR and aIL-10R). No effect on cell viability was observed (Supplementary Fig. 4a). The most prominent effect of LPS on DC hallmark maturation markers was observed at the mRNA level in the time frame of 4 to 8 h following activation[21]. We performed large-scale microarray analysis after 4 and 8 h of DC culture with LPS, with and without blocking Abs to TNF and IL-10 receptors (Fig. 5a).

Despite extensive studies of both TNF and IL-10 in the context of innate immunity, their different contribution to DC intercellular communication could not be predicted a priori at this systems level. We applied ICELLNET to reconstruct the intercellular networks between DCs and putative target cells. The network representation demonstrated an increase of the global communication score in all 12 channels, when comparing 8-hours LPS-activated DC to resting (medium) DC (Fig. 5b). Importantly, these maps revealed that blocking the IL-10 loop determined the largest amplification of DC communication with

all 12 cellular targets, while blocking the TNF loop in LPS-activated DCs had a negligible effect on the global communication score (Fig. 5b, Supplementary Fig. 4b and Supplementary Data 5).

**IL-10 controls an intercellular communication module in LPS-activated dendritic cells.** We compared the transcriptomic profiles of each condition (aTNFR and aIL-10R) to the LPS-alone condition to extract the differentially expressed genes (DEG) (Supplementary Data 5). We then screened the IL-10 and TNF DEG to identify ligands and receptors included in the database. We were able to extract 27 ligands and 23 receptors which were differentially regulated from the aIL-10R condition, while there were only 12 ligands and 10 receptors differentially regulated from the aTNFR condition (Fig. 5c).

ICELLNET barplots suggested that cytokines were driving the increase in the communication score when blocking IL-10R. We looked at the subfamilies of cytokines to precisely identify the key communication channels (Supplementary Fig. 4b) Type 1 and TNF subfamilies were increased in aIL-10R condition compared to others. This was confirmed by individual channel communication scores (Fig. 5d). To confirm the hypothesis that IL-10 controls cytokine-mediated DC communication, we selected four important immunoregulatory molecules from the IL-6- and IL-12-families, and further validated expression at the protein level in 24 h culture DC supernatants using cytometric bead array (CBA) and ELISA (Fig. 5e).

**Experimental validation of multiple IL-10-dependent communication channels.** To assess communication efficiency, i.e., how increased connectivity translates into functional changes in target cells, we turned to experimental validation of predicted communication channels using immunological assays adapted to the output response of each cell type. Due to its physiopathological relevance, we first investigated the DC-T cell axis through co-culture experiments of T cells with DCs treated by LPS with or without TNFR and IL-10R blocking antibodies (Fig. 6). We found that naive CD4$^+$ T cells, when co-cultured with LPS-DC in the absence of the IL-10 loop, globally increased and shifted their pattern of cytokine secretion, compared to LPS-DCs, while blocking the TNF loop had almost no effect (Fig. 6a). Similar results were obtained with memory T cells (Fig. 6b).

Since the IL-10/IL-10R pathway may have a direct effect on T helper cells during the differentiation process, we verified that the observed T helper polarization was indeed due to the IL-10 loop blockade in the DCs, and not due to a direct effect on T cells

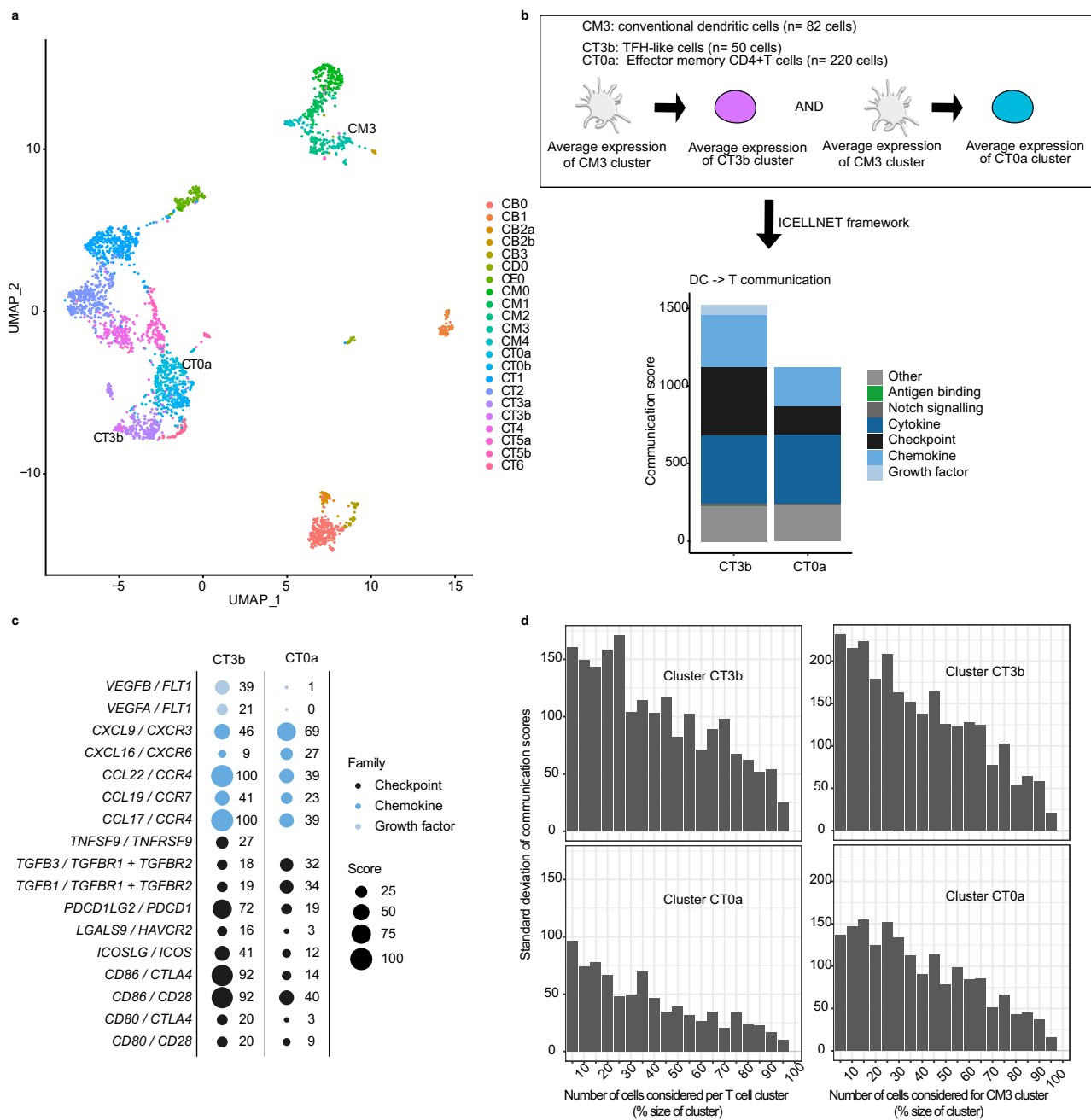

**Fig. 4 Evaluation of cell-to-cell communication potential between dendritic cells and T-cell subpopulations in lupus nephritis single cell data. a** Uniform Manifold Approximation and Projection (UMAP) visualization of the lupus nephritis dataset. 22 clusters were previously identified by the authors and their annotations are displayed on the right. Cell identity of each cluster can be found in the original article[20]. **b** ICELLNET framework applied on specific cluster to assess DC-T intercellular communication. Average expression profiles were computed from the single-cell data matrix counts for each cluster to then compute intercellular communication score. Barplots are displaying the contribution of the different communication molecules families to the communication scores. **c** Balloon plot representing specific individual interaction scores that differ from at least 10 between the two conditions (cutoff chosen for clarity purpose) for interaction belonging to either chemokine, checkpoint, or growth factor families of molecules. **d** Assessment of communication score variability when subsampling DC or T cluster. Cells were randomly selected for CT0a and CT3b cluster (left) or CM3 cluster (right) (number of cells selected according to their respective cluster size). Communication scores were computed with the other complete cluster (CM3 for left, T cell cluster for right). Standard deviations of the communication scores are displayed on the graphs (n = 20 random selection of cells). Some schematic art pieces were used and modified from Servier Medical Art, licensed under a Creative Common Attribution 3.0 Generic License. http://smart.servier.com/ (**b**).

(Supplementary Fig. 5a). It is possible that residual IL-10R blocking antibodies could have acted directly on T cells during DC-T co-culture. By adding IL-10R antibodies during DC-T co-culture (not only during DC activation) we demonstrated that any IL-10R antibodies in this setting would not have any direct effect on T-cell polarization.

Among the factors explaining the secretion profile of T cells determined by LPS + aIL-10R-DCs, we observed a remarkable emergence of Th17 cytokines (Fig. 6c), in line with murine studies[22,23]. Strikingly, IL-9 secretion was also increased (Fig. 6c), and produced by a T-cell population distinct from the Th17 cells producing IL-17A alone or co-expressed with IL-9 and IFN-g

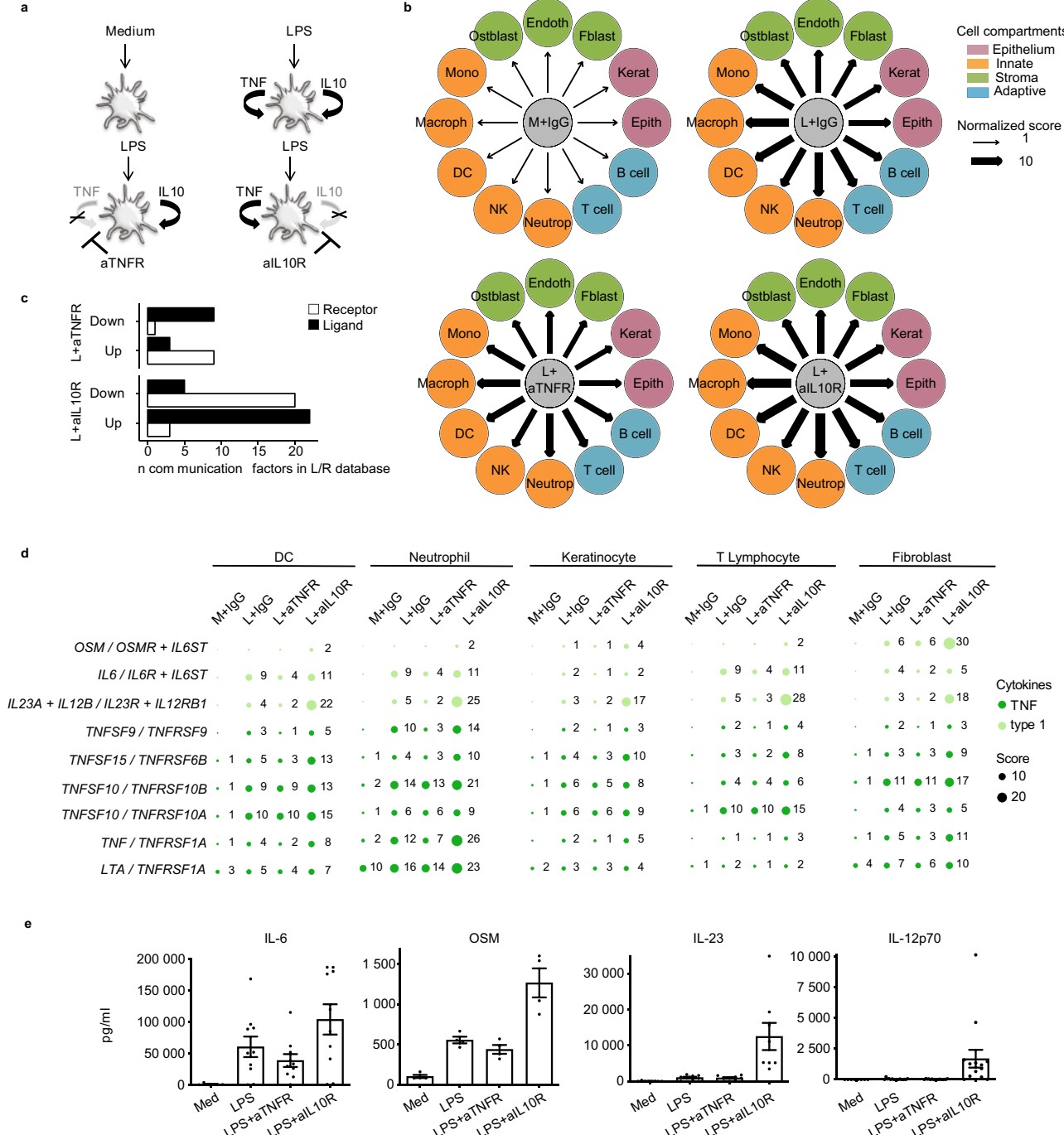

**Fig. 5 IL-10R blocking activates a cell-to-cell communication module in LPS-stimulated DCs. a** Depicted are the four experimental conditions for which transcriptomics was generated ($n = 6$ biologically independent samples). **b** Connectivity maps describing outward communication from DCs to putative target cells in the conditions: Medium (M), LPS (L), LPS + aTNFR and LPS + aIL-10R. Twelve primary cell types are considered as partner cells (DC: conventional dendritic cell, Macroph: macrophage, Mono: monocyte, Ostblast: osteoblast, Endoth: endothelial cell, Fblast: fibroblast, Kerat: keratinocyte, Epith: epithelial cell, B cell, T cell, Neutrop: neutrophil and NK: natural killer cell) and are colored depending on the cell compartment (green: stroma, orange: innate, blue: adaptive, pink: epithelium). The width of the edges corresponds to a global score combining the intensity of all the individual ligand/receptor interactions, normalized to the medium condition. A scale ranging from 1 to 10, corresponding to minimum and maximum communication scores, is shown in the legend. **c** Gene corresponding to ligands (black) and receptors (white) counted in each loop signature (from $n = 6$ biologically independent samples for each conditions) and plotted according to regulation directionality: upregulated (Up) or downregulated (Down). Genes with separability score $\geq 4$ were included in each condition's signature. **d** Balloon plot representing a selection of individual interaction scores. **e** Protein levels of IL-6 ($n = 10$ biologically independent samples, except for Medium condition where $n = 5$), OSM ($n = 4$ biologically independent samples), IL-23 ($n = 8$ biologically independent samples) and IL-12p70 ($n = 16$ biologically independent samples, except for Medium condition where $n = 8$) demonstrating increased secretion in LPS + aIL-10R DC supernatant. Data are presented as mean values ± SEM. Some schematic art pieces were used and modified from Servier Medical Art, licensed under a Creative Common Attribution 3.0 Generic License. http://smart.servier.com/ (**a**).

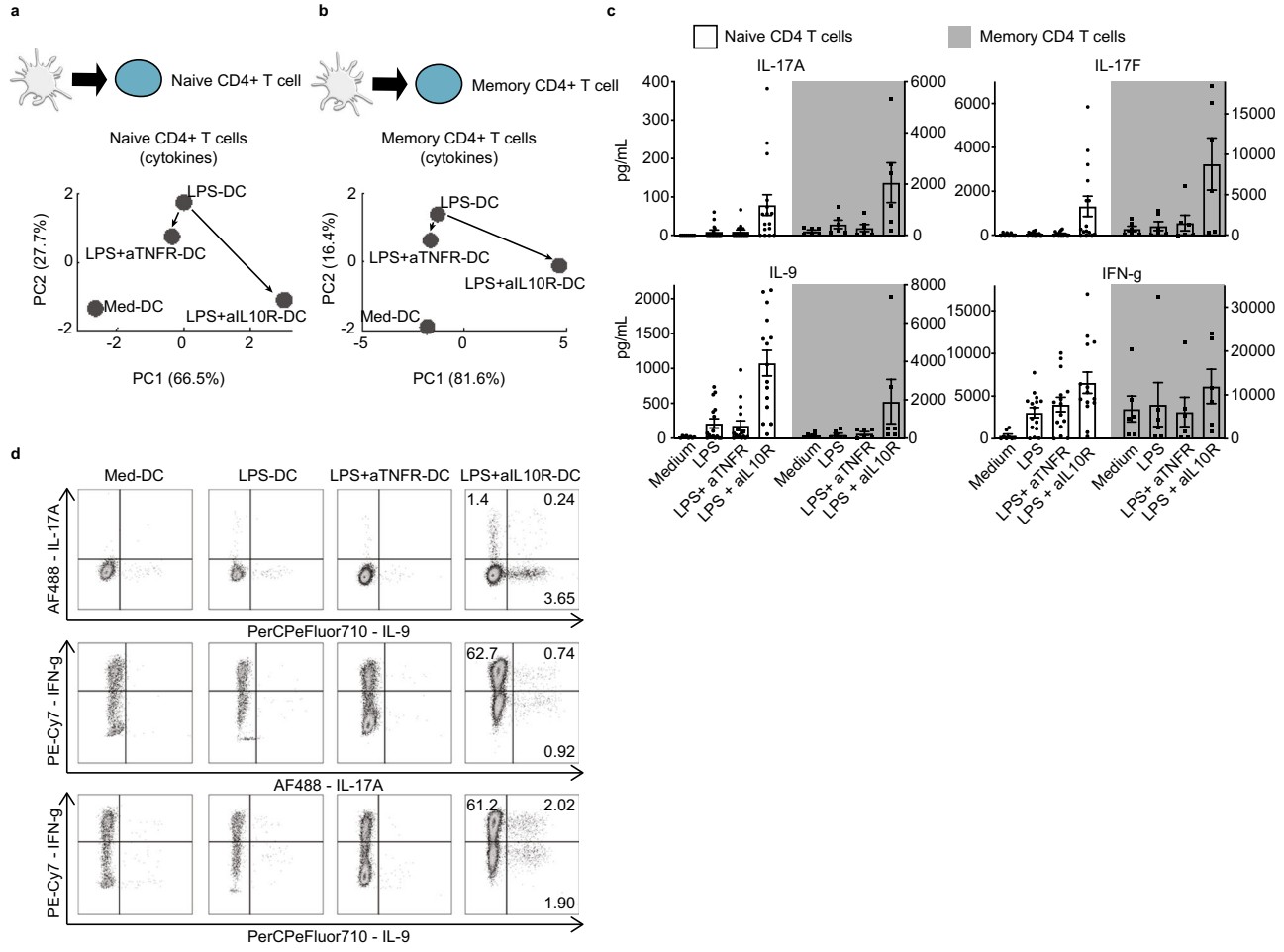

**Fig. 6 IL-10 but not TNF loop dictates T helper polarization by LPS-DC. a**, **b** Supernatants of CD4[+] naive (**a**) and memory (**b**) T cells, co-cultured with the indicated DCs, were analyzed for the presence of T helper cytokines by CBA: IL-2, IL-3, IL-4, IL-9, IL-10, IL-17A, IL-17F, and IFN-γ (**a**) and all the above in addition to IL-5, IL-13 TNF, and GM-CSF (**b**). Results are shown in a two-dimensional principal component analysis (PCA). Dots represent mean of 9 (**a**) or 6 (**b**) independent co-culture experiments. **c** Histogram representation of 4 cytokines present in the supernatant of naive (white bars, left axis) or memory (black bars, right axis) supernatant (n = 16 biologically independent samples for naïve CD4 T cells except Medium condition where n = 8 biologically independent samples, n = 6 for memory CD4 T cells). Data are presented as mean values ± SEM. **d** CD4[+] naive T cells were analyzed for IL-17A, IL-9, and IFNg production using intracellular staining FACS. Percentage of positive producers is given. Shown is one representative out of 3 independent experiments. Some schematic art pieces were used and modified from Servier Medical Art, licensed under a Creative Common Attribution 3.0 Generic License. http://smart.servier.com/ (**a**).

(Fig. 6d). This provides the demonstration that LPS-activated DCs, in the absence of an IL-10 loop, determine Th17 and Th9 polarization in humans, both of which participate in host defense and autoimmunity[24,25].

In order to validate the model-based hypothesis that there is increased communication between DC and multiple cell types, we considered three additional types of target cells: keratinocytes, plasmacytoid DCs (pDC), and neutrophils. Similar to T cells, these cell types play key roles in the inflammatory microenvironment and had an increased global communication score. Target cells were cultured with DC-derived supernatants, and their activation assessed by qRT-PCR or FACS. LPS-DC supernatant induced marginal keratinocyte activation, as assessed by the expression of TNF, IL-1β and this was not affected by aTNFR (Fig. 7a). However, blocking the IL-10 loop dramatically increased both factors (Fig. 7a), validating a potent DC to keratinocyte communication controlled by IL-10. This extends DC-induced keratinocyte activation[26,27] to the context of bacterial infection.

The DC-pDC communication channel was also controlled by IL-10, since LPS + aIL-10R-DC supernatants activated pDCs (as

assessed by CD86, HLA-DR, and ICOSL surface expression), in comparison to LPS-DCs (Fig. 7b). DC-induced activation of pDC and keratinocytes was not due to the presence of residual aIL-10R (Supplementary Fig. 5b, c). DC-pDC crosstalk was suggested to be important in antiviral[28], antibacterial[29], and antitumor[30] immune responses. Through our systems approach, we have shown that IL-10 controls DC-pDC connectivity.

Neutrophils contribute to DC migration to infection sites and to their subsequent activation[31,32]. Reciprocally, it was proposed that DCs can promote neutrophil survival[33]. LPS-DC supernatant induced only a mild activation of neutrophils (as evaluated by rapid upregulation of CD11b with concomitant downregulation of CD62L), while LPS + aIL-10R-DC supernatants led to a strong activation of neutrophils (Fig. 7c), establishing an IL-10 loop control of DC-neutrophils communication.

For all the abovementioned communication channels, we aimed at getting further mechanistic insight. First, we performed control experiments using exogenous LPS that formally excluded any direct effect of LPS at the concentrations found in the DC supernatants (Supplementary Fig. 5d). We then considered

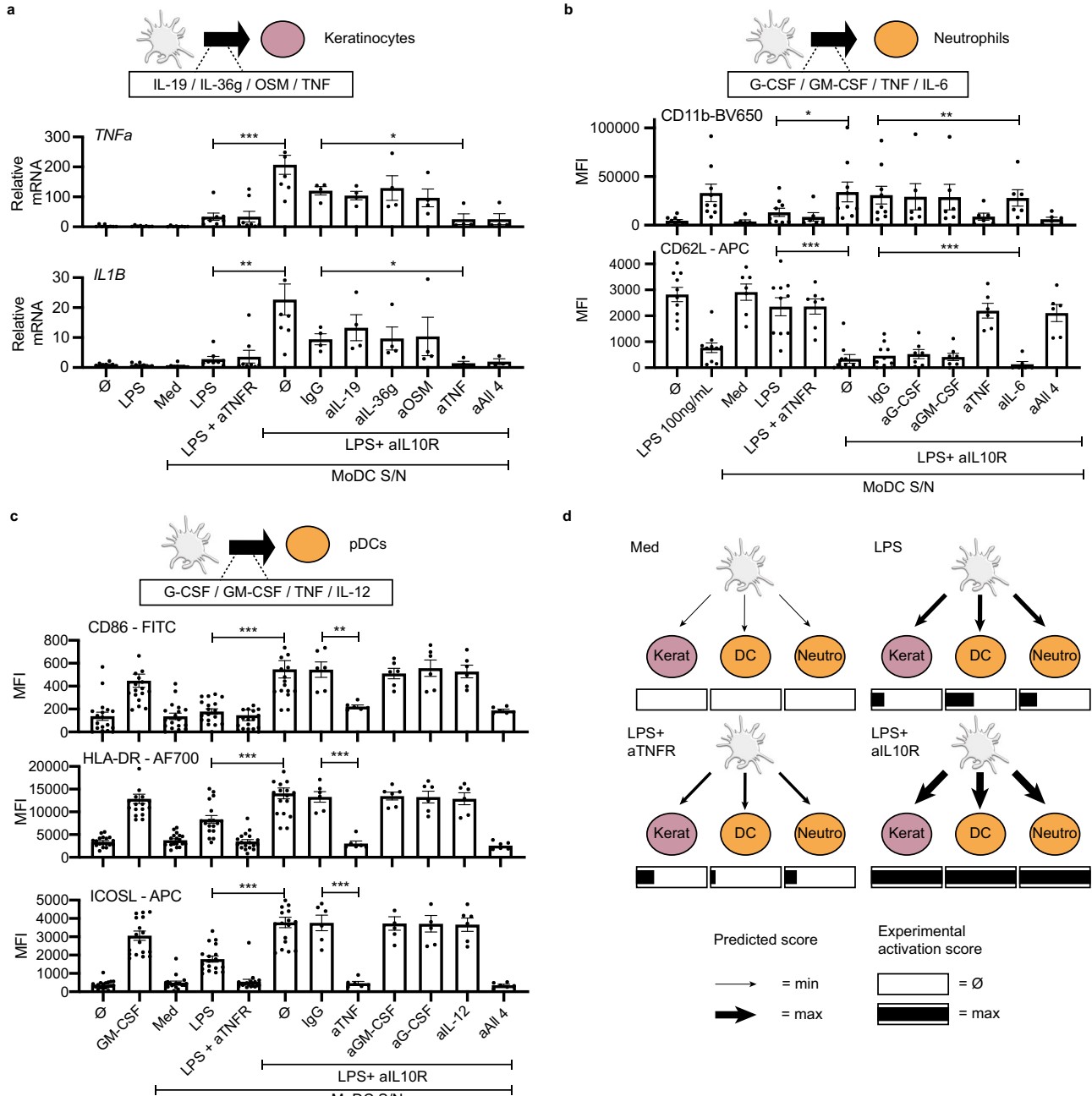

**Fig. 7 IL-10 loop controls DC communication with keratinocytes, neutrophils, and pDCs. a** RT-PCR analysis of the expression of TNF and IL-1b mRNA in HaCat cells incubated with medium, LPS or with supernatant (diluted 1:10) of the indicated DCs for 4 h. Blocking antibodies for the cytokines IL-19, IL-36g, OSM and TNF were added to LPS + aIL-10R-DC supernatant for 1-h incubation before culturing with HaCat cells. Data are represented as mean values ± SEM, $n = 4$ or $n = 8$ biologically independent samples depending on the conditions, *$p < 0.05$ (two-sided paired $t$-test without correction). **b, c** Expression of maturation markers CD86, HLA-DR, and ICOSL (**b**) or CD11b and CD62L (**c**) analyzed by flow cytometry with surface staining on pDCs ($n = 18$ or $n = 6$ biologically independent samples depending on the conditions) cultured with supernatant (diluted 1:10) of the indicated DC for 24 h (**b**) and neutrophils ($n = 9$ or $n = 6$ biologically independent samples depending on the conditions) (**c**) cultured with supernatant (diluted 1:100) of the indicated DC for 1 h. Blocking antibodies for the cytokines GCSF, GM-CSF, TNF, and IL-12 (for pDC) or IL-6 (neutrophils) were added to LPS + aIL-10R-DC supernatant for 1 h incubation before culture. Each biological replicate comprised independent DC donor paired to independent pDCs/neutrophils donor. Data are represented as mean values ± SEM, *$p < 0.05$; **$p < 0.01$; ***$p < 0.001$ (two-sided paired $t$-test without correction). **d** For each target cell, we reduced the different activation markers to a single parameter normalized between 0 (Ø) and 1 (max) in the rectangles. The value 0 corresponds to the activation level induced by supernatants from untreated DC, while 1 corresponds to the maximum activation level from all the observed conditions. These experimentally validated activation scores were in qualitative agreement with the ICELLNET communication scores between DC and the target cells, represented by the width of the edges. Some schematic art pieces were used and modified from Servier Medical Art, licensed under a Creative Common Attribution 3.0 Generic License. http://smart.servier.com/ (**a**, **b**, **c**).

ligand–receptor interactions showing high intensity, and thus more likely to mediate cellular crosstalk as observed with the LPS + aIL-10R-DC supernatants (Supplementary Data 6). We blocked, in each DC communication channel, four of the ligands known as potential activators of the target cell type: GCSF, GM-CSF, IL-6, and TNF for neutrophils, IL-19, IL-36 gamma, OSM and TNF for keratinocytes, and G-CSF, GM-CSF, TNF and IL-12 for pDCs. Importantly, blocking TNF alone in the LPS + aIL-10R-DC supernatant was sufficient to inhibit keratinocyte, pDC and neutrophil activation (Fig. 7a–c). By comparing the predicted communication intensities with a global score describing the activation level of keratinocytes, pDC and neutrophils, we observed a qualitative agreement (Fig. 7d), demonstrating increased communication efficiency. In all cases, the target cells were most activated in LPS + aIL10R condition.

## Discussion

The majority of studies which aim to reconstruct intercellular communication from transcriptomic datasets integrate prior knowledge in the form of a ligand–receptor interaction database. This provides a straightforward manner to infer communication when a match is identified between a ligand and a cognate receptor for two respective cell types. The largest of such databases[9] integrated over 2500 ligand–receptor pairs through literature mining and computational analysis, and has been exploited in multiple computational tools for predicting cell-to-cell communication[5,8,34,35]. However, this approach lacks experimental validation of predicted ligand–receptor interactions, and it does not take into account the different subunits of ligands or receptors. With ICELLNET, we have developed a fully manually curated database, combining biological relevance, ease of use, and experimental validation. Except for one study[5], ICELLNET is the only database offering a classification of predicted interactions into biological families. Similar to CellPhoneDB[10], ICELLNET takes into account the multiple subunits of ligands and receptors, by introducing logical rules for co-expression of protein subunits. A systematic comparison of cytokine interactions revealed 14 interactions included in ICELLNET but not in CellPhoneDB, such as *MIF/CXCR2* and *MIF/CXCR4*[36]. Although ICELLNET includes a relatively small number of interactions compared to other existing databases, it is very specific and exhaustive for cytokine interactions, and will in time be extended to all chemokine and checkpoint interactions, thus providing a unique resource to study intercellular communication within the immune system.

To make ICELLNET a valuable resource, we have established a strategy to keep the database updated and integrate missing knowledge. A significant number of interactions have been established in the past 20 years, but there are still receptors without known ligands, such as DR6 (*TNFRSF21*), RELT, TROY and NGFR from TNF receptor family[37], and ligands without known receptors such as IL17D[16]. New receptors for existing ligand–receptor pairs can also be uncovered using this approach. For example, even though it was already known that IL34 and M-CSF could separately activate M-CSFR[38] it was then described that IL34/M-CSF heterodimer was also capable of activating M-CSFR[39]. We will apply a PubMed alert strategy to cover all new interactions that could be described on these pre-identified ligand and/or receptor partners.

Existing tools infer communication between cells from single-cell RNA-seq datasets[4,5,7]. We have designed ICELLNET as a versatile tool, which can be applied to bulk cell profiles (Affymetrix or RNA-seq) widely available in public databases, but also to fully documented single-cell RNA-seq datasets to infer communication between clusters or groups of cells. This could be also easily adapted to other types of data such as flow cytometry data.

By using the Human Primary Cell Atlas as a reference for transcriptomic profiles[17,18], ICELLNET allows us to integrate cell communication partners (sender or receiver) not included in a given original experimental dataset. We identified the Human Primary Cell Atlas as a particularly suitable resource, as it integrates transcriptional profiles of over thirty human primary cell types generated with the same Affymetrix platform[18]. While previous applications of this atlas enabled the identification of specific tissue-related genes[40,41], we developed an original use for this resource to simulate cell cross-talks in diverse microenvironments. In addition, ICELLNET can accommodate other original RNA-seq datasets of cell populations[42,43] as reference profiles to infer intercellular communication. Hence, ICELLNET is a flexible tool, which can be easily adapted depending on the biological question, by offering the possibility to select communication molecules families and cell types of interest.

Interesting approaches that integrate downstream signaling pathways and target gene expression have been explored as a surrogate of an effective communication[7,44]. Even if very relevant, these approaches can lead to false positive and false negative predictions, the frequency of which is almost impossible to predict. Redundancy and promiscuity in signaling pathways can lead to false positive results, for example due to co-expressed receptors triggering the same canonical pathway. Conversely, functional interactions, due to the possible presence of numerous communication molecules targeting the same cell, may perturb the downstream canonical pathways, leading the false negative results, i.e., an effect not attributed to a given ligand–receptor interaction, when such interaction effectively occurs. We have previously established the multimodality in interactions between two stimuli, which makes their prediction almost impossible given the current state of knowledge in downstream signaling pathways[45]. Due to these limitations, we have decided not to integrate downstream effectors in ICELLNET, at this stage. This may evolve as our understanding of interactions between stimuli improves.

A key aim in studying cell–cell communication is to represent cellular interactions in a clear and biologically relevant manner. Visualization is important to understand the different levels of interactions, at the cellular and molecular levels. Most of the available tools use two main graphical representations; heatmaps and circos plots. These complex plots represent all possible interactions at once and can be difficult to read and interpret. ICELLNET offers four original visualization modes with different properties to represent cell-to-cell communication from a global view of specific interactions. These different representations simplify interpretation of the results, help users to elaborate hypotheses, and allow in-depth analysis of cell-to-cell interactions.

The cytokine family of communication molecules plays a key role in homeostatic processes, such as cell development and differentiation, tissue homeostasis, and inflammation[3,14,46]. In the past 20 years, a large number of new cytokines have been identified, cloned, and studied to elucidate their biological function. This has significantly enriched the classification of cytokines into structural families matching evolution and functional processes[3,47]. ICELLNET is now providing an exhaustive and expert curated resource of all known cytokines and their receptor interactions, according to reference knowledge. This opens possibilities for researchers to decipher complex cytokine-mediated communication, and the implication of specific cytokines in disease.

Fibroblasts are important structural stromal cells at steady state and inflammation. Yet, how they communicate with neighboring cells is not well described. Applying ICELLNET to breast cancer fibroblasts' bulk cell transcripts revealed potentially novel interactions between CAF subsets and tumor microenvironment components. The *CXCL12/CXCR4* interaction that we found within

CAF-S1-to-Tregs (Fig. 3d) was also described in other studies[19,48], and contributes to the immunosuppressive phenotype displayed by CAF-S1. ICELLNET also highlighted interactions specific to CAF-S4 subset such as JAG1 with Notch receptors (NOTCH1, NOTCH2), and expression of PDGF proteins interacting with their cognate receptor. These proteins have never been associated specifically to CAF-S4 subset at the transcriptomic level and warrant further experimental validation studies.

Other studies have shown that IL-10 regulates DC-derived inflammatory cytokines and chemokines, in particular IL-12[49], and that IL-10 secreted by LPS-activated DC controls a communication channel in an autocrine manner[50]. Through our systems approach, we could demonstrate that endogenous DC-derived IL-10 governs the global connectivity of DC with multiple cell types. This original in vitro dataset generated in a controlled system allowed us to experimentally validate four intercellular communication channels, which established the robustness and biological relevance of ICELLNET predictions. We did not find this level of experimental validation in any of the other cell–cell communication inference methods to date[5,10,35].

In conclusion, ICELLNET is an adaptable tool that allows us to gain insight into communication channels between cells from one bulk transcriptomic profile of a cell population. By focusing on specific cell types or families of molecules, ICELLNET provides several representation modes to help the interpretation of the results. Experimentally validated with an in vitro system, ICELLNET enables the dissection of intercellular communication in complex systems.

## Methods

**Human primary cell atlas dataset**. The dataset contains 745 samples of over thirty human primary cell types in different biological conditions (rested or activated). Included in BioGPS platform, all the samples have been generated with the same Affymetrix technology (Human Genome U133 Plus 2.0 arrays). For this study, an already processed and normalized dataset has been downloaded and added to ICELLNET package. This dataset is described in Supplementary Data 2.

**CAFs RNA-seq data processing**. The dataset contains 77 samples from Luminal (Lum) and Triple Negative Breast Cancers (TNBC) from 16 patients (10 Lum, 6 TNBC)[19]. The samples correspond either to tumor tissue or juxtatumoral tissue. Cells corresponding to CAF-S1 and CAF-S4 have been isolated, collected, and sequenced. Average sequencing depth was 30 million for paired-end reads, with a read length of 100 bp. Reads were mapped on the reference genome (hg19/GRCh37 from UCSC genome release) using Tophat_2.0.6 algorithm. Duplicates were removed and gene expression quantification was performed with HTSeq-Count and featuresCount. Only genes with five reads in at least 25% of all samples were kept for further analyses. Normalization was done using the method implemented in DESeq2 R package (version 1.26.0). In this study, only 6 biologically independent samples of CAF-S1 and 3 biologically independent samples of CAF-S4 from TNBC were considered in the analyses.

**Purification of peripheral blood mononuclear cells (PBMCs) from adult blood**. Fresh blood samples were collected from healthy donors and obtained from Hôpital Crozatier Établissement Français du Sang (EFS), Paris, France. A contract (convention) has been established between French Blood establishment (EFS) and Institut Curie, in conformity with national regulations and ethical guidelines. Written informed consent was obtained for each healthy donors. PBMCs were isolated by centrifugation on a Ficoll gradient (Ficoll-Paque PLUS, GE Healthcare Life Sciences).

**Monocyte-derived dendritic cells generation and activation**. Monocytes were selected from PBMCs using antibody-coated magnetic beads and magnetic columns according to manufacturer's instructions (CD14 MicroBeads, MiltenyiBiotec). To generate immature DCs, CD14+ cells were cultured for 5 days with IL-4 (50 ng/mL) and GM-CSF (10 ng/mL) in RPMI 1640 Medium, GlutaMAX (Life Technologies) with 10% FCS. Monocyte-derived DCs were pre-treated for 1 h with mouse IgG1 (20 µg/mL, R&D Systems), mouse anti-IL-10R blocking antibody (10 µg/mL, R&D Systems) or mouse anti-TNFα Receptors 1 and 2 (10 µg/mL, R&D Systems) (Fig. 5 and Supplementary Fig. 4b) and then cultured with medium or LPS (100 ng/mL, LPS-EB Ultrapure, activates TLR4 only, Invivogen) for 24 h. DCs from donors which responded to (a) LPS and (b) IL-10R blocking antibody, as evaluated by maturation markers, were included in this study. The following

cytokines were measured in culture supernatants by CBA (BD Bioscience): IL-6, IL-12p70, and OSM. IL-23 was measured using ELISA (eBioscience).

**DC gene expression profiling**. Monocyte-derived DCs were pre-treated with blocking Abs as described above for 1 h and then cultured with medium or LPS (100 ng/mL, Invivogen) for an additional 4 or 8 h. Total RNA was extracted using the RNeasy micro kit (Qiagen). Samples were then amplified and labeled according to the protocol recommended by Affymetrix for hybridization to Human Genome U133 Plus 2.0 arrays. If multiple probes corresponded to the same receptor, we selected the optimal probe based on the Jetset optimality condition[51].

**Curation of the ligand/receptor database**. Surveying the literature for any potential interactions, we manually curated a ligand–receptor database using STRING (https://string-db.org/), Ingenuity (https://www.ingenuity.com/), and BioGRID (https://thebiogrid.org) online tools to verify protein–protein interactions, as well as Reactome (https://reactome.org) and CellPhoneDB (https://www.cellphonedb.org) databases, already dedicated to ligand–receptor interactions.

As robustness of an interaction, we considered interactions described in at least two independent published resources (reviews, original papers, existing ligand–receptor database), written by different authors for reviews and original articles. Consistency criterion was used to compare interactions described in different resources, checking that same protein subunits are involved. "Experimental validation" criterion was in particular useful to check for specific interactions with few descriptions or no consistency between resources. Original articles demonstrating the interaction were reported in PubMedID column of the database.

The interactions were classified into families of molecules based on the known biological function of the ligand and the receptor. The subfamilies of cytokines were defined based on molecular structures, as defined in the literature[3,14–16]. The database of ligand–receptor interactions is contained in the Supplementary Data 1.

**Gene expression matrix scaling method**. After selecting the genes corresponding to the ligands and/or receptors from the transcriptional profiles, each ligand/receptor gene expression is scaled by maximum of gene expression among all the conditions and then multiplied by 10, to have values ranging from 0 to 10. For each gene, the maximum value (10) is defined as the mean of expression of the 5% highest values of expression for RNA-seq and microarray datasets. Outliers are rescaled at 10 if above maximum value.

**Intercellular communication score computation**. To score the intensity of a particular ligand–receptor interaction between a central cell and a given partner cell, we considered the product of the expression of the ligand in the central cell and of the cognate receptor in the partner cells. Formally, if $l_j^i$ is the average expression level of ligand $i$ by the central cell in the experimental condition $j$, and $r_k^i$ is the average expression of the corresponding receptor by cell type $k$, the intensity $s_{j,k}^i$ of the corresponding interaction was quantified by $s_{j,k}^i = l_j^i \cdot r_k^i$. For interactions requiring multiple components of the ligand and/or of the receptor, we considered a geometric average of the receptor components. For example, if a given interaction corresponding to ligand $i$ required two chains of the receptor, the score was computed as $l_j^i \cdot \sqrt{r_k^{i,1} \cdot r_k^{i,2}}$, where $r_k^{i,1}$ and $r_k^{i,2}$ are the expression levels of the two receptor chains in cell type $k$. To assign a global score $S_{j,k}$ to the communication between the central cell in the condition $j$ and cell type $k$, a composite score was defined by summing up the intensity of all the possible ligand–receptor interactions, i.e., $S_{j,k} = \sum_{i=1}^{N} s_{j,k}^i$, $N$ being the total number of interactions. If there is any technical effect in one dataset (central cell or partner cell) it is considered as a weight coefficient $w$:

$$S_{j,k} = \times \sum_{i=1}^{N} w_j^i l_j^i w_k^i r_k^i \qquad (1)$$

$$S_{j,k} = w \times \sum_{i=1}^{N} s_{j,k}^i \qquad (2)$$

This weight is common for all cells selected from the dataset of central or partner cell and can be considered as a multiplication factor, so it only affects the range of the communication score but not the relative difference between them. Regarding the four DC experimental conditions (Medium ($j = 0$), LPS ($j = 1$), blocking TNF loop ($j = 2$), blocking IL-10 loop), we normalized the global scores $S_{j,k}$ to the Medium condition ($j = 0$) across the four conditions. Thus, the final scores $\overline{S_{j,k}}$ used to measure the communication intensity between DC in the condition $j$ and the target cell $k$ were computed using the following formula:

$$\overline{S_{j,k}} = S_{j,k}/S_{0,k} \qquad (3)$$

$$\overline{S_{j,k}} = \frac{\sum_{i=1}^{N} s_{j,k}^i}{\sum_{i=1}^{N} s_{0,k}^i} \qquad (4)$$

The score corresponding to each interaction and each target cell in the experimental condition of CAF subsets, lupus nephritis patients single-cell

RNA-seq dataset, and the four DC experimental conditions are provided in Supplementary Data 3, 4, and 5 respectively. The generation of the inward connectivity maps was done by reversing the role of the central cell and their cellular targets.

**Global intercellular communication score scaling method**. The intercellular communication scores are rescaled ranging from 1 to 10, considering all the scores computed for each biological condition between the central cell and all selected partner cell types. This step allows us to increase the differences between the scores and facilitate the network visualization of the communication scores.

**Statistical comparison of communication scores**. To compare the communication scores obtained from the same central cell to different partner cells we compute several communication scores considering the average expression of ligands for the central cell and each replicate separately for the receptor expression of the partner cells. In this way, for one partner cell type, we obtain a distribution of $n$ communication scores, $n$ being the number of partner cells replicates for this particular cell type. Second, we can compare communication scores between two biological conditions. In this case, we compute several communication scores considering each replicates of the central cell separately, and the average gene expression for the partner cells. We obtain a distribution of $n$ communication scores, $n$ being the number of central cell replicates in one biological condition. For both cases, we then perform a Wilcoxon statistical test to compare the communication scores distributions. The $p$-values are adjusted with p.adjust() function from the R package « stats » (version 3.6.1) using the Benjamini & Hochberg[52] method in R. This returns the $p$-value matrix of statistical tests, that can be visualized in a heatmap representation with the pvalue.plot() function from « icellnet » R package.

**Statistical analysis of gene expression data**. Expression data were normalized with Plier. Transcriptomics analysis was performed in Matlab (version R2010). For independent filtering, we used the function geneverfilter, which calculates the variance of each probe across the samples and identifies the ones with low variance. Probes with variance less than the 40th percentile were filtered out. Differential analysis was performed using an ANOVA test (function anova1) at 4 h and 8 h. $p$-values were adjusted for multiple testing using the Benjamini–Hochberg correction using the function mafdr. Adjusted $p$-values < 5% were considered significant (see Supplementary Data 5).

**Purification of naive CD4$^+$ T lymphocytes**. CD4$^+$ T lymphocytes were purified from PBMCs by immunomagnetic depletion with the human CD4$^+$ T cell Isolation KitII (MiltenyiBiotec), followed by staining with allophyco-cyanin-anti CD4 (VIT4; MiltenyiBiotec, dilution 1:40), phycoerythrin-anti-CD45RA (BD, dilution 1:20), fluorescein-isothiocyanate-anti-CD45RO (BD, dilution 1:20), and phycoerythrin-7-anti-CD25 (BD, dilution 1:20). Naive CD4$^+$ T cells sorting of CD4$^+$CD45RA$^+$CD45RO$^-$CD25$^-$ and memory CD4$^+$ T cells sorted as CD4$^+$CD45RA$^-$CD45RO$^+$CD25$^-$ had a purity of over 99% with a FACSAria (BD Bioscience). A representative gating strategy is provided in Supplementary Fig. 6a.

**DC–T cells coculture assays**. To analyze T-cell polarization, 24 h activated DC and T cells were incubated in 96 well plates at a DC/T ratio 1:5 in Xvivo15 medium (Lonza). After 6 days, T cells were resuspended in fresh Xvivo15 medium at a concentration of 1 million cells per mL and restimulated with anti-CD3/CD28 beads (life Technologies) at a ratio bead/cell 1:1. Supernatants of T cells were collected after 24 h of restimulation. The following cytokines were measured in naive culture supernatants by Cytometric Bead Array (CBA) (BD Bioscience) according to the manufacturer's instructions: IL-2, IL-3, IL-4, IL-9, IL-10, IL-17A, IL-17F, and IFN-γ. Additional cytokines were measured in memory T cells supernatant: IL-5, IL-13, TNF, and GM-CSF. Cytokine-producing cells were analyzed by intracellular staining after addition of brefeldinA (10ug/mL) during the last 3 h of the 5 h restimulation in PMA and ionomycine (100 ng/mL and 500 ng/ml, respectively). Cells were stained for 30 min with the yellow live dead kit (Invitrogen). Finally, cells were fixed and permeabilized using the Staining Buffer Set (eBiosciences) and stained with PerCPeFluor710 anti-IL9 (eBiosciences, 1:20), Pe-Cy7 anti-IFNg (eBiosciences, 1:50), and anti-IL17A (Biolegend, 1:80), and analyzed by flow cytometry (BD Fortesa).

**Measurement of surface molecules expression by plasmacytoid dendritic cells**. In order to enrich plasmacytoid dendritic cells (pDCs), cells expressing CD3, CD9, CD14, CD16, CD19, CD34, CD56, CD66b, and glycophorin A were depleted from PBMCs using magnetic sorting (Human Pan-DC Pre-Enrichment Kit, StemCell Technologies). pDCs were then sorted on a FACS Vantage instrument (BD Biosciences). A representative gating strategy is provided in Supplementary Fig. 6b. pDCs were cultured for 24 h at 37 °C and 5% CO$_2$ with medium RPMI 1640 Medium, GlutaMAX (Life Technologies) with 10% FCS, GM-CSF (10 ng/mL) used as a positive control or DC supernatants. Cells were stained for 15 min at 4 °C using a FITC-anti-CD86 (BD, 1:20), an APC-anti-ICOSL (R&D Systems, 1:20) and

Alexa-Fluor-700-anti-HLA-DR (Biolegend, 1:20) or with the corresponding isotypes. Cells were analyzed on an LSR II instrument (BD Biosciences).

**Measurement of adhesion molecules expression at the neutrophil surface**. PBMCs were stimulated for an hour at 37 °C with medium, LPS (100 ng/mL) used as a positive control or DC supernatants. Cells were stained at 4 °C for 15 min with an APC-anti-Human-CD62L (clone DREG-56, BD Pharmingen, 1:50), a BV650-anti-Human-CD11b (BioLegend, 1:120) and a PE-anti-Human-CD15 (BD Pharmingen, 1:20) or with the corresponding isotypes. Erythrocytes were lysed with 1X BD Pharm Lyse Solution (BD Pharmingen). White cells were resuspended in PBS supplemented with 1% human serum and 2 mM EDTA and analyzed on an LSR Fortessa instrument (BD Biosciences).

**Real-time quantitative RT-PCR**. The keratinocyte cell line HaCaT was kindly provided by Prof. Dr. Bernhard Homey. HaCaT cell line was cultured in DMEM (Gibco) supplemented with 10% FBS and 1% penicillin/streptomycin. Cells were cultured with medium, LPS (100 ng/ml), or with DC supernatant diluted 1:10 for 4 h. Total RNA was extracted by RNeasy Mini kit (Qiagen). RNA was then transcribed to cDNA using Superscript II reverse transcriptase based on the manufacture's protocol (Invitrogen). The Taqman method was used for real-time PCR with primers from Life technologies (Supplementary Table 1). The expression of mRNA was normalized to the geometrical mean of 3 house-keeping genes: ACTB, GAPDH, and RPL34. All HaCaT cells were negative for Mycoplasma contamination, standardized and regular tests were performed by PCR for mycoplasma detection.

**Statistical analysis of DC-T-cell protein data**. All analyses were generated with R 3.1 or 3.6.3. For principal component analysis (PCA) of the T-cell secretion profile, a data matrix was formed whose rows corresponded to conditions and columns to the different cytokines (each column was scaled using zscore). PCA was done using the function princomp. Where appropriate, a two-sided paired student $t$-test was performed. Significant differences were considered with $p < 0.05$.

**Calculation of the activation score of target cells**. To compute a global activation score of keratinocytes, neutrophils and pDC, each activation marker output was first normalized in the range 0–1, 0 being to the untreated condition and 1 being to the maximum value observed in all the conditions. An average of the normalized outputs corresponding to the same cell type was then considered. All of the measured factors, with the exception of CD62L in neutrophils, were positively correlated with cell activation. In order to make CD62L consistent with the other factors, we considered the reciprocal of its value. The numerical results are in the Supplementary Data 6.

**Reporting summary**. Further information on research design is available in the Nature Research Reporting Summary linked to this article.

## Data availability

The gene expression profiles generated for this publication have been deposited in NCBI's Gene Expression Omnibus and are accessible through GEO Series accession number GSE89342.

The CAFs dataset has been published by Costa et al.[19], and is accessible through the accession number EGAS00001002508. The single cell dataset of immune cells from lupus nephritis patients has been published by Arazi et al.[20], and is accessible through the ImmPort repository (accession code SDY997).

## Code availability

ICELLNET package is available at https://github.com/soumelis-lab/ICELLNET and has been deposited in Zenodo (https://doi.org/10.5281/zenodo.4327491)[53].

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

## Acknowledgements

We wish to thank Melissa Saichi, Lilith Faucheux, Gaël Blivet-Bailly, Fanny Coffin, Philippe Hupé, Franck Perez, Sebastian Amigorena, Yong-Jun Liu for insightful comments and discussions, and Fivos Soumelis for conceptual advice on the definition, functions, and formalization of communication in living systems. We thank the Institut Curie Flow Cytometry facility (Z. Maciorowsky), and the Institut Curie Affymetrix facility (D. Gentien). This work was supported by funding from Institut National de la Santé et la Recherche Médicale (Inserm), the Institut Curie, Agence Nationale pour la Recherche (ANR), Fondation pour la Recherche Médicale (FRM), the European Research Council (ERC starting grant 281987), ANR-10-IDEX-0001-02 PSL, ANR-11-LABX-0043, CIC IGR-Curie 1428 for V.S. EMBO, Institut Curie post-doctoral fellowships to ICL. ANRS and ARC fellowships to MG and La Ligue Nationale Contre le Cancer doctoral fellowship to LMR.

## Author contributions

F.N. and L.M.-R. designed and performed bioinformatic analyses and wrote the manuscript. I.C-L. designed and performed experiments and analyzed results. A.C performed bioinformatic analyses. M.G. and C.T. performed experiments and revised the manuscript. Y.K. and F.M.-G. provided strategic advices and revised the manuscript. V.S. designed experiments, supervised the research, and wrote the manuscript. I.C.L. and A.C contributed equally to this work.

## Competing interests

Irit Carmi-Levy is a full-time employee at Aummune. Maximilien Grandclaudon is currently employed by the pharmaceutical company Servier. The remaining authors declare no competing interests.

**Additional information**

