## [Peer Review File · Nature Communications]

Reviewers' comments:

Reviewer #1 (Remarks to the Author):

Noël et al. present a computational method to predict cell-cell communication networks based on single-cell RNA sequencing data, through development of a curated database of ligand-receptor interactions and a metric to quantify communication between cells. Better tools to infer relationships between cells based on single-cell profiling are certainly needed, and this work offers several appealing features, including helpful visualizations and a well-presented R package. However, I have a number of major concerns that limit the potential of these methods for community use and scientific discovery. These are listed below.

Major points

1. The authors present “an original expert- curated database of ligand-receptor interactions accounting for multiple subunits expression” - the inclusion of receptor subunits is indeed useful, but no detailed description of how the expert curation was performed (other than “manual curation”) is given, limiting its utility.
2. The authors offer “quantification of communication scores” - the metric used (taking the product of expression of ligand and receptor) is simplistic and subject to fluctuation, e.g. differences between lowly and highly expressed ligands/receptors will be amplified. The scaling used might mitigate this, but assessment of this score is needed. It is of concern that the score does not take into account target gene expression: ligand-receptor interactions, while helpful, cannot provide sufficient information to support cell-cell communication interactions, since in many cases receptors are expressed widely and not indicative of pathway activation. Two recent works have taken target gene expression into account (PMID: 30923815, PMID: 31819264) - these should be discussed, and most likely incorporated into the scoring metric for ICELLNET in order to make useful predictions about cell-cell communication.
3. The focus on ICELLNET on sums over ligand-receptor pairs is confusing, and goes against the premise of an “expert curated database”, i.e. one for which the results of particular ligand-receptor pairs are interesting and meaningful. Studying signaling within specific pathways, for example mediated by a family of ligands (e.g. WNTs or BMPs) is indeed useful, but it is hard to unravel what is gained by grouping together “cytokines” - an incredibly diverse group, some of which certainly will have opposing effects on some cell populations (thus confounding ICELLNET predictions). In several places, the result of, e.g., “more communication” is stated. This seems near-meaningless when summing over these many different signaling pathways.
4. I am concerned by how summing over different cells and cell types is treated. Very little discussion of either of these points is given. Little discussion of the advantages and challenges of working with single-cell data is given. How do the networks handle variability between cells? What strategies are implemented to handle the particular sources of noise in these data? In addition, results are presented for which different datasets from different experiments are combined. Due to the known prevalence of batch effects in single-cell RNA sequencing data, it seems like great caution must be exercised when combining data in such a way, and the authors do not demonstrate that this has been handled or even thought about in the work. (e.g. the CAFs with the cell atlas data)
5. The authors claim that ICELLNET works with both bulk or single-cell data — not enough support for this is given. These two sources of RNA sequencing have particular differences; making it hard to see where this method should be applied. I.e. does it make use of any of the unique features of single-cell data? Given the grouping of cells into populations, it seems perhaps not? The concept of the “central cell” is helpful but needs to be supported - it is not clear how the central cell is calculated. Is it simply a cell cluster? The greatest finding to come out of widespread investigations into the transcriptomics of single cells is the high levels of variability between cells. Simply summing over cells does not account for this.

6. The authors make claims that I don't think can be supported, in particular, that ICELLNET has been experimentally validated (claim made in several places) and that other alternative databases "lacks experimental validation" (pg. 19). One cannot say that an entire method has been experimentally validated if a few (or even many) of its predicted interactions have been tested experimentally. More careful language must be used: "out of XXXX predictions made by this method using XXXX dataset, XXXX have been tested experimentally, and...."

Minor points

Fig 2 - not well labeled as "S4_TN" is used but not explained (does not apply to this figure).

Fig 3D - legend misleading - why choice to plot nothing for < 15? How is this cutoff chosen? Similar for Fig 4D.

Reviewer #2 (Remarks to the Author):

The authors present a computational framework called ICELLNET that facilitates the prediction of cell-cell communication based on transcriptome, and particularly single-cell RNA sequencing data. First, the authors present an original, expert-curated data base of ligand-receptor interaction. I especially like the detailed annotation of respective family, subtypes and presentation format (soluble or membrane-bound) of the ligand-receptor pairs.

Then, the authors present ICELLNET methodology and an experimental validation of their finding using two case studies that illustrate how the approach can be used for research. While the framework is well designed and appears to be able to deliver robust analysis, I would not consider any of the concepts significantly novel. The general concepts and ideas have been published before, however, the authors present here a robust and valuable implementation that I think will advance the field. Also, the case studies are well worked out and the experimental validation is convincing. As such, those case studies are interesting by themselves.

The article is well written and presented in high quality, the literature review and reference is thorough and facilitates judgment of the scientific advancement this manuscript provides. I support the publication and have some minor comments.

Minor:

- Fig 2, I understand this should explain the approach in general. However, there seem to be some specific labels "S4_TN" and "S1_TN" that are not explained. It appears those are somewhat related to the later application of Fig 3. To avoid confusion, I would use easily understandable terms in this figure (e.g. CAF-subtype1).

- Page 6, you introduce the notation of "peripheral cells" here, as I understand those are cells elsewhere in the body. I find this notation could lead to confusion with PBMCs. Maybe another description can be found that is less confounded?

- Fig 3: Again, there are labels "S4_TN" and "S1_TN" here in panels B) and E) that are not clear. I would keep notation consistent in the figure and call those "CAF S1" and "CAF S2" throughout. What does the "_TN" stand for?

- Figure legends 1 and 2 lack details. Please elaborate more on what the figures show in details.

Reviewers' comments:

Reviewer #1:

Noël et al. present a computational method to predict cell-cell communication networks based on single-cell RNA sequencing data, through development of a curated database of ligand-receptor interactions and a metric to quantify communication between cells. Better tools to infer relationships between cells based on single-cell profiling are certainly needed, and this work offers several appealing features, including helpful visualizations and a well-presented R package. However, I have a number of major concerns that limit the potential of these methods for community use and scientific discovery. These are listed below.

Major points

- 1. The authors present “an original expert- curated database of ligand-receptor interactions accounting for multiple subunits expression” - the inclusion of receptor subunits is indeed useful, but no detailed description of how the expert curation was performed (other than “manual curation”) is given, limiting its utility.**

REPLY 1: We have revised the manuscript (page 4, Results section first paragraph) to explain with more clarity and details how our ligand-receptor database has been generated. The manual curation is based on rigorous literature screening of articles with biological insight and consensus reviews written by specialists recognized in their field by the scientific community. Thus, we included in the database interactions that were robust, reproducible, consistent, and experimentally validated. In order to provide an up-to-date and reliable classification of each molecule into a molecular family, we used our own expert knowledge of cytokine- and chemokine-mediated communication, as well as published consensus reviews, providing reference and shared knowledge (for example, Zlotnik et al. 2012, Dinarello et al. 2010, O’Shea et al. 2019). Whenever appropriate, we took into account the multiple subunits of the ligands and receptors. We did not include putative interactions based on protein-protein interaction predictions, as it is done in some other databases such as Ramilowski et al 2015.

For each interaction, we reported the PubMed ID (PMID) of the original article describing the interaction, or of consensus reviews for well-established interactions. Reviews were also used to define the classification of interactions by family of molecules, and the subfamilies of cytokines based on the structure of receptors and cytokines functions. Finally, members of our team with expertise in immunology and cytokines checked the interactions included in the database.

To sum up, this database is the first of our knowledge to include both the multiple subunits of ligand and receptors, and a classification into families/subfamilies of the interactions, using expert curation and reference publication in each specific field.

- 2. The authors offer “quantification of communication scores” - the metric used (taking the product of expression of ligand and receptor) is simplistic and subject to fluctuation, e.g. differences between lowly and highly expressed ligands/receptors will be amplified. The scaling used might mitigate this, but assessment of this score is needed.**

REPLY 2: We believe that the metric used (taking the product of expression of ligand and receptor) is the easiest and yet efficient method to assess the expression of both the ligand and receptor on the cell types of interest, also taking into account the relative gene expression level intensity. However, a gene expression scaling is needed. All molecules do not have the same bioactivity, which means that they do not require the same level of expression to be biologically active. This aspect is also more importantly reflected at the transcriptomic level. For example, IL12 is highly bioactive and is often expressed at low levels, whereas chemokines are less bioactive and generally highly expressed. We have explained this in the revised version of the manuscript (page 7, Results section third paragraph and page 21, Methods section). Since the reviewer was concerned about amplification of differences between lowly and highly expressed ligand/receptor, we slightly modified the scaling method in the revised version of the manuscript. For each gene we normalized the expression by its maximum value across the samples and multiplied by 10. This scaled the gene expression between 0 and 10 taking into account the intrinsic range but not the raw intensity.

It is of concern that the score does not take into account target gene expression: ligand-receptor interactions, while helpful, cannot provide sufficient information to support cell-cell communication interactions, since in many cases receptors are expressed widely and not indicative of pathway activation. Two recent works have taken target gene expression into account (PMID:30923815, PMID:31819264) - these should be discussed, and most likely incorporated into the scoring metric for ICELNET in order to make useful predictions about cell-cell communication.

REPLY 3: As the reviewer mentioned, there are already published interesting approaches that account for downstream pathways and target gene expression to predict cell-cell communications. We believe that these methods are very prone to false positive and false negative predictions, at a rate that is unpredictable, and highly dependent on the nature of the communication molecules, and the complexity of the microenvironment they may be subjected to. This implies that the false positive and false negative rates would be very variable. Indeed, the redundancy and the promiscuity in signaling pathways can lead to false positive results, due for example to co-expressed receptors triggering the same canonical pathway. Conversely, false negative results in the case of a particular functional interaction, could be due to the presence of another communication molecule that could modify or perturb the canonical pathway on the same target cell, leading to emergence of a new effect of the interaction or unknown effects.

We added a paragraph to discuss this part in the revised manuscript (page 17, Discussion section). We have cited one of our previous studies that established the concept of multimodality in signal integration, also pointing at unpredictability of interactions between stimuli (Cappuccio et al., 2015). However, we recognized the relevance of this question, while acknowledging its complexity. Reaching sufficient knowledge in interactions between communication molecules may require several years of work by the scientific community. In the current state of knowledge, we prioritized a simple, robust and versatile method based on prediction of ligand-receptor interactions. The fact that we experimentally validated a number of predictions gives us confidence in its robustness and biological relevance.

3. The focus on ICELLNET on sums over ligand-receptor pairs is confusing, and goes against the premise of an “expert curated database”, i.e. one for which the results of particular ligand-receptor pairs are interesting and meaningful. Studying signaling within specific pathways, for example mediated by a family of ligands (e.g. WNTs or BMPs) is indeed useful, but it is hard to unravel what is gained by grouping together “cytokines” - an incredibly diverse group, some of which certainly will have opposing effects on some cell populations (thus confounding ICELLNET predictions). In several places, the result of, e.g., “more communication” is stated. This seems near-meaningless when summing over these many different signaling pathways.

REPLY 4: Our ligand-receptor interactions database includes knowledge on individual molecules but also families of molecules, based on the most up-to-date international classifications. These classifications often take into account the molecular structure and the function of the molecules to define families and subfamilies of molecules. ICELLNET tool offers graphical representations at different levels of granularity, including 1) a global view of the communication, 2) a graphical representation at a level of family of molecules, 3) subfamilies of molecules (for cytokines), and 4) a last graphical representation showing specific ligand-receptor pairs.

The comment of the reviewer refers to the first and second analytical levels, which may have some limitations but are powerful to rapidly assess intensity and type of communication between two cells. The utility of grouping communication molecules into large families may vary between types of scientific question, as well as cell types, and physiopathological conditions. However, ICELLNET also includes two additional analytical levels (3 and 4) which move towards increasing granularity in the analyses. The versatility of ICELLNET as a framework allows the user to very easily decide at which level the analysis should be done.

As for the functions and output effects of these communication molecules, we agree with the reviewer that they are extremely diverse, but also pleiotropic and highly context-dependent. This makes it almost impossible to group them by function, and this may be the reason current classifications are based on structure and high level functionalities, rather than specific functions. We designed ICELLNET as a predictor and hypothesis-generator on the type and intensity of the communication between cell types, but not on the functional output of the communication, which would require specific experimental studies. In this way we aim at minimizing confounding predictions.

4. I am concerned by how summing over different cells and cell types is treated. Very little discussion of either of these points is given.

REPLY 5: Regarding the different cell types we advise the users to use cell types data coming from the same dataset for the partner cells and/or for the central cells. It makes it easier for the pre-processing of the data prior to ICELLNET framework usage. In our manuscript, partner cells data (or central cells data) are processed following the same pipeline in order to limit batch effects. The communication score corresponds to the sum of the product of ligand-receptors pairs expression. If there is any technical effect in one dataset (central cell or partner cell) it will be considered as a weight coefficient w : $\text{Sum}(\text{Li} * w_c * \text{Ri} * w_p) = (w_p * w_c) * \text{Sum}(\text{Li} * \text{Ri})$. This weight is common to all cells

selected from the dataset of central or partner cell so it can be considered as a multiplication factor. It will only affect the range of the communication score but not the relative difference between them. To clarify this aspect, we better explained the calculation of the score in the revised manuscript (page 22, Methods section).

Little discussion of the advantages and challenges of working with single-cell data is given. How do the networks handle variability between cells? What strategies are implemented to handle the particular sources of noise in these data?

REPLY 6: Variability between cells is indeed an inherent challenge of any “single cell” study. ICELLNET was not designed to overcome the problem of variability between cells in single cell data. The user chooses the granularity of cell grouping (during clustering) and then ICELLNET will use the average expression of communication molecules for each cluster. However, following the reviewer’s question, we added a result section on ICELLNET application to single-cell RNA sequencing data (revised manuscript page 11, and new Figure 4) (see also reply 8 below). In this analysis, we assessed the robustness of ICELLNET predictions to random subsampling of single cells clusters. We found that ICELLNET was robust to subsampling. However, ICELLNET communication scores variability was anti-correlated with the number of cells selected for the subsampling. In other words, predicting communication on a 3-cell cluster led to different results than a 100-cell cluster due to the fact that a 3-cell cluster does not integrate all the biological characteristics of the original cluster. This reflects the inherent variability of single cells, and is now quantified in the revised version of our manuscript.

In addition, results are presented for which different datasets from different experiments are combined. Due to the known prevalence of batch effects in single-cell RNA sequencing data, it seems like great caution must be exercised when combining data in such a way, and the authors do not demonstrate that this has been handled or even thought about in the work. (e.g. the CAFs with the cell atlas data)

REPLY 7: As described in Reply 5, we have clarified the manuscript to better describe how we handle batch effect (Methods section, page 23) and why it does not affect ICELLNET communication scores and analyses.

5. The authors claim that ICELLNET works with both bulk or single-cell data – not enough support for this is given. These two sources of RNA sequencing have particular differences; making it hard to see where this method should be applied. I.e. does it make use of any of the unique features of single-cell data? Given the grouping of cells into populations, it seems perhaps not?

REPLY 8: ICELLNET has been developed to compute a communication score between two cell types based on their transcriptomic profiles, independently of the number of cells included. Thus, it can be applied to bulk RNA-seq data from cell population, but also to single-cell RNAseq data. In this case, as the reviewer mentioned it, ICELLNET is indeed preferentially applied to groups of

cells. This still offers some versatility as the level of granularity of cell cluster is defined by the user. We do not recommend to use ICELLNET on individual cells: this is not a limitation of the computational method per se, but a limitation of current single cell RNAseq technologies which include sparsity and dropout in the source data. Thus, we believe that the analyses obtained with individual cells would not be robust and biologically relevant.

Following the reviewer's comments, we have added a case study on single cell data in the revised version of the manuscript to discuss these aspects, and explain with more clarity how ICELLNET framework should be applied on this type of data (revised manuscript pages 10-11, and new Figure 4).

The concept of the “central cell” is helpful but needs to be supported - it is not clear how the central cell is calculated. Is it simply a cell cluster?

REPLY 9: The central cell is not “calculated”; it is defined by the user depending on the cell type of interest from which the user wants to infer cell-cell communication networks. Once this is defined, ICELLNET will use the transcriptomic profile of the central cell (either from Human Cell Atlas if the user has decided so, or other specific transcriptomic profiles already loaded in R by the user) to compute intercellular communication score. In the case of single-cell data, the central cell can be defined as the average transcriptomic profile of a cell cluster.

We have modified the main text to explain this with more clarity (revised manuscript page 6, Results section).

The greatest finding to come out of widespread investigations into the transcriptomics of single cells is the high levels of variability between cells. Simply summing over cells does not account for this.

REPLY 10: Again, as mentioned in reply 8, there are inherent limitations in single-cell RNAseq methodologies that prevent from using individual cells in order to study their communication potential (i.e. sparsity and dropout). ICELLNET could in theory be applied to individual cells, and could give interesting results on communication potential but technologies need to be improved in order to be robust and biologically relevant at a level of individual cells.

6. The authors make claims that I don't think can be supported, in particular, that ICELLNET has been experimentally validated (claim made in several places) and that other alternative databases “lacks experimental validation” (pg. 19). One cannot say that an entire method has been experimentally validated if a few (or even many) of its predicted interactions have been tested experimentally.

REPLY 11: This is an interesting point and a challenge in any systems biology study. At the era of “omics” data and computational biology, we are facing situations where tens or even hundreds of predictions may come out of a systems approach. Experimental validation is costly and lengthy, which creates a potential bottleneck. For this reason, experimentally validating a large number of predictions is not feasible. Most systems biology studies end up validating one prediction, in order to support a whole strategy (Jerby-Arnon et al, 2018 ; Gilchrist et al, 2006. Sometimes experimental validation cannot be addressed because of cell number limitations, not decreasing the interest of

the work (Michea et al, 2018). In our current study, we could experimentally validate testable hypotheses generated in a controlled system. We experimentally tested four communication channels predicted between DC and other cell types, and all four were experimentally validated. This is way above average when considering other systems biology studies, and was not performed in other studies of cell communication using systems levels inference by other methods. We consider it a very important aspect supporting the robustness of our method, and the biological relevance of the predictions.

More careful language must be used: “out of XXXX predictions made by this method using XXXX dataset, XXXX have been tested experimentally, and....”

REPLY12: We thank the reviewer for his remark and we improved the text in the manuscript to clarify that we experimentally validated four communication channels that were predicted by ICELLNET framework.

Minor points

Fig 2 - not well labeled as “S4_TN” is used but not explained (does not apply to this figure).

REPLY 13: We thank the reviewer for noticing this misleading label and we modified the figure.

Fig 3D - legend misleading - why choice to plot nothing for < 15? How is this cutoff chosen? Similar for Fig 4D.

REPLY 14: In this graphical representation, all individual pairs are displayed as long as their contribution to the score is above a cutoff value. The cutoff was chosen at 15 in the previous figure 3 for clarity. But this value is not always the relevant cutoff, and is defined by the user according to the number of individual pairs he would like to see on the graph. This allow to rapidly identify the individual pairs that contribute the most to the communication score. In the new figure 3, the cutoff was set at 10 after the modification of the gene expression scaling method.

We thank the reviewer for these comments and suggestions, which helped us clarify and improve the quality of our article. We incorporated all the suggested changes.

Reviewer #2 (Remarks to the Author):

The authors present a computational framework called ICELLNET that facilitates the prediction of cell-cell communication based on transcriptome, and particularly single-cell RNA sequencing data. First, the authors present an original, expert-curated data base of ligand-receptor interaction. I especially like the detailed annotation of respective family, subtypes and presentation format (soluble or membrane-bound) of the ligand-receptor pairs.

Then, the authors present ICELLNET methodology and an experimental validation of their finding using two case studies that illustrate how the approach can use used for research. While the framework is well designed and appears to be able to deliver robust analysis, I would not consider any of the concepts significantly novel. The general concepts and ideas have been published before, however, the authors present here a robust and valuable implementation that I think will advance the field. Also, the case studies are well worked out and the experimental validation is convincing. As such, those case studies are interesting by themselves.

The article is well written and presented in high quality, the literature review and reference is thorough and facilitates judgment of the scientific advancement this manuscript provides. I support the publication and have some minor comments.

Minor:

- Fig 2, I understand this should explain the approach in general. However, there seem to be some specific labels "S4_TN" and "S1_TN" that are not explained. It appears those are somewhat related to the later application of Fig 3. To avoid confusion, I would use easily understandable terms in this figure (e.g. CAF-subtype1).

REPLY 15: We thank the reviewer for the comment, this has been modified in the revised version of the manuscript.

- Page 6, you introduce the notation of "peripheral cells" here, as I understand those are cells elsewhere in the body. I find this notation could lead to confusion with PBMCs. Maybe another description can be found that is less confounded?

REPLY 16: We thank the reviewer for this comment. To avoid confusion with blood peripheral primary cells, we have changed the name into « partner cells » to highlight the fact that these cells are interacting with the central cell.

- Fig 3: Again, there are labels "S4_TN" and "S1_TN" here in panels B) and E) that are not clear. I would keep notation consistent in the figure and call those "CAF S1" and "CAF S2" throughout. What does the "_TN" stand for?

REPLY 17: TN stands for Triple Negative Breast Cancer. As the reviewer mentioned, this abbreviation is not clear and adds no value to the figure. We then decided to remove it and has been removed for more clarity in the revised version of the manuscript. CAF-S1 and CAF-S4 stand for the 2 subsets of cancer-associated fibroblasts that have been identified by Costa et al, 2018. Thus, we will keep their annotations to avoid any confusion with the other CAFs subsets.

- Figure legends 1 and 2 lack details. Please elaborate more on what the figures show in details.

REPLY 18: We thank the reviewer to have noticed this. We explained the figure legends with more details in the revised version of the manuscript (page 32).

We thank the reviewer for these comments and suggestions, which will help us to clarify and improve the quality of our article. We will incorporate all the changes suggested.

REVIEWERS' COMMENTS

Reviewer #2 (Remarks to the Author):

All my questions have been addressed sufficiently, no further comments.

Reviewer #3 (Remarks to the Author):

The ICELLNET application will be useful to folks in the immunology community. The application is limited in scope and this should be stated up front. Some of the claims are overstated and not fully justified for example

1. Abstract

"ICELLNET notably revealed autocrine IL-10 as a switch to control human dendritic cell communication with up to 12 other cell types, four of which were experimentally validated. In summary, ICELLNET is a global, versatile, biologically validated, and easy-to-use framework to dissect cell communication from individual or multiple cell-based transcriptomic profile(s)."

Please tone down conclusion - a 4/12 validation for one ligand is a 33% success rate and it is not reasonable then to call the application global and versatile

2. Results

"An expert manual curation was performed based on a rigorous literature screening of original articles, applying the following criteria: 1) robustness of the findings, 2) consistency with international classifications and nomenclature, 3) reproducibility, 4) experimental validation of the functionality of the ligand receptor interaction. We also used consensus reviews from leaders in the field, in particular for cytokines."

For these criteria how is robustness of findings defined - multiple papers by different groups, multiple papers by same authors? Also what is the difference between the robustness of findings and reproducibility. The authors probably recognize the important of these assertions as they underline them . But they need to describe better how these criteria are implemented perhaps with examples in the Supplementary Materials

3. Why was IL-10 chosen as an example – please describe rationale

Point by point – Final revisions – ICELLNET

Reviewer #2 (Remarks to the Author):

All my questions have been addressed sufficiently, no further comments.

Reviewer #3 (Remarks to the Author):

The ICELLNET application will be useful to folks in the immunology community. The application is limited in scope and this should be stated up front. Some of the claims are overstated and not fully justified for example

1. Abstract

"ICELLNET notably revealed autocrine IL-10 as a switch to control human dendritic cell communication with up to 12 other cell types, four of which were experimentally validated. In summary, ICELLNET is a global, versatile, biologically validated, and easy-to-use framework to dissect cell communication from individual or multiple cell-based transcriptomic profile(s)."

Please tone down conclusion - a 4/12 validation for one ligand is a 33% success rate and it is not reasonable then to call the application global and versatile

Reply: We believe this part of the abstract was misunderstood. Actually, IL-10 was predicted to control dendritic cell communication with up to 12 other cell types. Out of those 12 cell types, 4 were selected for further testing and experimental validation. All 4 were successfully validated: results matched the predictions. To clarify this, we modified the abstract:

"ICELLNET reveal autocrine IL-10 control of human dendritic cell communication with up to 12 cell types. Four of them (T cells, keratinocytes, neutrophils, pDC) are further tested and experimentally validated."

2. Results

"An expert manual curation was performed based on a rigorous literature screening of original articles, applying the following criteria: 1) robustness of the findings, 2) consistency with international classifications and nomenclature, 3) reproducibility, 4) experimental validation of the functionality of the ligand receptor interaction. We also used consensus reviews from leaders in the field, in particular for cytokines."

For these criteria how is robustness of findings defined - multiple papers by different groups, multiple papers by same authors? Also what is the difference between the robustness of findings and reproducibility. The authors probably recognize the important of these assertions as they underline them. But they need to describe better how these criteria are implemented perhaps with examples in the Supplementary Materials

Reply: We recognize that robustness of the finding and reproducibility are confusing because very similar meaning. Thus, and to avoid ambiguity, we decided to remove the "reproducibility" term from the manuscript that for us is included in the "robustness of the finding" criteria. As suggested by the reviewer, each criterion has been explicitated in the

method section (page 21 of the manuscript), in order to give more details on the ligand-receptor pairs curation procedure:

“As “robustness of an interaction”, we considered interactions described in at least two independent published resources (reviews, original papers, existing ligand-receptor database), written by different authors for reviews and original articles. “Consistency” criterion was used to compare interactions described in different resources, checking that same protein subunits are involved. “Experimental validation” criterion was particular useful to check for specific interactions with few descriptions or no consistency between resources. Original articles demonstrating the interaction were reported in PubMedID column of the database.”

3. Why was IL-10 chosen as an example – please describe rationale

Reply: This point was clarified in the revised manuscript, results section, page 12: “LPS-activated human monocyte-derived DC, produce two important autocrine cytokines, TNF and IL-10, which play a major role in regulating inflammation. We asked whether this could be mediated by modulating DC communication with partner cells.”

The rationale and physiopathological relevance is also addressed in the discussion of the original manuscript.